# Homogeneity criteria from AVHRR information within IASI pixels in a Numerical Weather Prediction context

Imane Farouk[1], Nadia Fourrié[1], and Vincent Guidard[1]

[1]CNRM UMR 3589, Université de Toulouse, Météo-France, CNRS, Toulouse, France

**Correspondence:** Nadia Fourrié (nadia.fourrie@meteo.fr)

IASI, homogeneous scenes, clouds, ARPEGE model

**Abstract.** This article focuses on a selection of satellite infra-red IASI observations and their simulation in the global Numerical Weather Prediction (NWP) system ARPEGE (Action de Recherche Petite Echelle Grande Echelle), using the sophisticated radiative transfer model RTTOV-CLD which takes into account the multi-layer clouds and the cloud scattering from atmospheric profiles and cloudy microphysical parameters (liquid water content, ice content and cloud fraction). The aim of this work is to select homogeneous scenes by using information of the collocated Advanced Very High Resolution Radiometer (AVHRR) pixels inside each IASI field of view and to retain the most favourable cases for the assimilation of IASI infrared radiances. Two methods to select homogeneous scenes using homogeneity criteria already proposed in the literature were employed: criteria derived from Martinet et al. (2013) for cloudy sky selection in the French mesoscale model AROME (Applications of Research to Operations at MEsoscale), and the criteria from Eresmaa (2014) for clear sky selection in the global model IFS (Integrated Forecasting System). A comparison between these methods reveals considerable differences, both in the method to compute the criteria and in the statistical results. From this comparison a revised method is proposed that is a compromise between the different tested methods, using the two infrared AVHRR channels to define the homogeneity criteria in the brightness temperature space. This revised method has a positive impact on the observation minus the simulation statistics, while retaining 36% observations for the assimilation. It was then tested in the NWP system ARPEGE and tested for the clear-sky assimilation. These criteria were added to the current data selection based on the Mc Nally and Watts (2003) cloud detection. It appears that the impact on analyses and forecast is rather neutral.

## 1 Introduction

Satellite observations are currently the dominant source of information for Numerical Weather Prediction (NWP) systems. Their assimilation together with in-situ observations give the atmosphere analysis, which is a necessary step in the definition of the initial conditions of the forecast. This analysis consists in finding a state of the atmosphere that is compatible with the different sources of observations, the dynamics of the atmosphere and a previous state of the model. In the Météo-France global model ARPEGE (Action de Recherche Petite Echelle Grande Echelle, Courtier et al. (1991)), 70% of used observations come from infrared hyperspectral sounders, of which IASI (Infrared Atmospheric Sounding Interferometer, Cayla (2001)) fills a large part. This sounder provides information about the atmospheric temperature and humidity, and through its window

channels information about the land surface parameters in clear sky and cloudy parameters can be obtained. However, the wealth of information provided by this type of sensor with its large number of channels or radiances (8461 per pixel in the case of IASI) and its overall coverage with a horizontal resolution of 12 km at nadir, is far from being fully exploited. Indeed, the presence of clouds in the instrument field of view, which affects the majority of observations, prevents from an accurate

simulation of the radiances. In fact, NWP centres use only a small amount of observations from these sounders mostly in clear sky above clouds. Previous studies have shown that sensitive areas are often covered by clouds (McNally (2002), Fourrié and Rabier (2004)) and different techniques have been developed in the frame of global models to use infrared radiances in these regions.

In the past, different approaches have been proposed for cloud detection. A method to detect clear channels from high-
resolution InfraRed (IR) spectral instruments was proposed by McNally and Watts (2003) to assimilate channels unaffected by clouds.

At the Met Office, Pavelin et al. (2008) showed that it was possible to assimilate cloud-affected infrared radiances when retrieved cloud parameters are used as set constraints. The cloud-top pressure (CTOP) and the effective cloud fraction (Ne) are firstly retrieved by a one-dimensional variational data assimilation system (1D-Var) and then provided to four-dimensional
variational data assimilation (4D-Var) for the assimilation of cloud-affected infrared radiances. The analysis is significantly improved over the first guess by this method and it is used operationally to assimilate AIRS and IASI cloud-affected radiances. At ECMWF (European Centre for Medium-Range Weather Forecasts), McNally (2009) proposed a method based on two cloud parameters (CTOP and Ne) to assimilate cloud-affected IR radiances directly. In that case, the cloud parameters are determined with two channels and they are then introduced into the analysis control vector of the 4D-Var system of the global NWP model
to constrain the minimization.

At Météo-France, the cloud parameters (CTOP and Ne) are retrieved for AIRS and IASI cloud-affected radiances with the $CO_2$-slicing method Menzel et al. (1983). Channels affected by clouds, the cloud top pressure (CTOP) of ranges from 650 to 900 hPa with an effective cloud fraction (Ne) of 1, are assimilated in addition to clear ones in the ARPEGE 4D-Var and the AROME (Applications of Research to Operations at MEsoscale) 3D-Var Pangaud et al. (2009) and Guidard et al. (2011).

As pointed out by Errico et al. (2007), studies on the assimilation of clouds and precipitation from satellite sensors have started in the 80s and despite the encountered difficulties to implement them, operational weather centres are now assimilating them with a clear benefit for the forecast quality. Efforts started with microwave radiances and direct all-sky microwave radiance assimilation is effective at ECMWF (Bauer et al., 2010) since 2009 and at NOAA (National Oceanic and Atmospheric Administration) NCEP (National Centers for Environmental Prediction) since 2016. Even though ECMWF focussed on the as-
similation of microwave imaging and humidity-sounding channels and NOAA NCEP, on the contrary, of temperature channels from Advanced Microwave Sounding Unit-A (AMSU-A), both centres noticed benefits of such an all-sky assimilation on the forecast quality (Geer et al., 2017; Zhu et al., 2016).

Concerning infrared radiance all-sky assimilation, no operational centre is yet to assimilate infra-red observations but research has still started in this area. Many aspects have already been studied as the information on cloud microphysics brought
by the adjoint sensitivity in the assimilation (Greenwald et al., 2002) or by the retrieval of cloudy infrared radiances (Martinet

et al., 2013). In addition the sensitivity, the reproducibility and the nonlinearity of the simulation of IR radiances in the presence of multi-layer clouds was studied using diagnosed cloud schemes (Chevallier et al., 2004) and (Stengel et al., 2010). These studies also showed beneficial results.

A step further was achieved with the study by Okamoto et al. (2014). They studied the assimilation of multi-layer cloud-affected infrared radiances using the all-sky assimilation approach already implemented for microwave imager at ECMWF. They particularly investigated the cloud effects on the differences between observations and simulations and thus proposed an appropriate quality check and dedicated observation errors.

In this study we are interested in IASI observations, where the radiances are considered with colocated clusters statistical properties of the Advanced Very High Resolution Radiometer (AVHRR) also on board the Metop platform. AVHRR has a horizontal resolution of 1km at nadir (Cayla, 2001). Intuitively, collocated AVHRR data provide information on surface properties and the presence of clouds in the IASI Field Of View (FOV). They can therefore be used for cloud detection. The AVHRR cluster information associated with IASI has already proven to be useful for selection purposes in the context of cloud data assimilation, with an explicit treatment of microphysical variables in the AROME model by Martinet et al. (2013). Eresmaa (2014) at ECMWF also used AVHRR cluster information for cloud detection and observation selection in clear sky.

Martinet et al. (2013) selected cloudy scenes based on cloud homogeneity. This study was done in a 1D-Var framework using an advanced radiative transfer model (RTTOV-CLD) including profiles for liquid water content, ice water content and cloud fraction to simulate cloud-affected radiances as background equivalents to AROME fields. The persistence of cloud information brought by the analysis of cloud variables during a 3h forecast has then been evaluated successfully with an one-dimensional model AROME version (Martinet et al., 2014). Okamoto (2017) studied the impact of the super-observation homogeneity quality control on the Advanced Himawari Imager brightness temperature simulation. He concluded that for larger size of super-observations, the standard deviation threshold should be relaxed in order to keep sufficiently low brightness temperatures associated with high-level cloud because of the presence of more cloud heterogeneity in large size observations.

In this article, our objective is to determine homogeneity criteria valid for both clear and cloudy conditions, suitable to a NWP context using collocated AVHRR and IASI information. we try to determine whether or not collocated AVHRR and IASI information would facilitate the selection of homogeneous scenes which could be potentially used in an all sky assimilation approach. Section 2 describes the ARPEGE NWP system, the IASI instrument and the radiative transfer model RTTOV-CLD in cloudy sky conditions. In section 3, information about the AVHRR clusters is detailed, the strengths and weaknesses of the different methods to select homogeneous observations are discussed, the chosen method is presented together with a description of the selected observations. Section 4 depicts the impacts on analyses and forecasts of selected clear and cloudy IASI observations. Conclusion and perspectives are given in section 5.

## 2   Experimental framework

### 2.1   The ARPEGE model and its 4D-Var system

The ARPEGE model is the global NWP model at Météo-France, used operationally since the early 1990s (Courtier et al., 1991). This system is fully integrated within the ARPEGE-IFS software that was conceived, developed and maintained in collaboration with ECMWF.

This model is a spectral global model with a stretched grid having a horizontal resolution around 7.5 km over France and 37 km over the antipodes. It has 105 vertical levels according a following-terrain pressure hybrid coordinate, with the first level located at 10 m above the surface and an upper level at around 70 km. Clouds and precipitation are described by using three different scheme in the ARPEGE model. The stratiform clouds in terms of cloud profile and precipitation are explicitly modeled from the microphysical condensation scheme by Lopez (2002). The large-scale effects of deep convection are parametrized from a mass-flux scheme derived from Bougeault (1985) and the shallow convection ones with the Bechtold et al. (2001) one. In these last two cases, the cloud fraction and the liquid water, ice and precipitation profiles are diagnosed.

ARPEGE has four analyses per day at 00, 06, 12 and 18 UTC. Since June 20th, 2000 the operational data assimilation system of the ARPEGE model is a 4D-Var. This implementation, as detailed in Janiskova et al. (1999) and Rabier et al. (2000), is used to provide an analysis which corresponds to the best atmospheric state knowing observations, an a-priori state, dynamical and physical constraints.

The background errors are computed at each analysis time based on the 25-member assimilation ensemble (see Berre et al. (2015) for further details). The control variables considered are temperature, specific humidity, vorticity, diverence and the logarithm of the surface pressure.

At each analysis around 7 million observations are assimilated. They include conventional observations (from radiosounding, aircraft, ground stations, ships, buoys, etc.) and satellite data. These latter include radiances in the infrared and microwave spectra such as AIRS (Atmospheric InfraRed Sounder), IASI, CrIS (Cross-track Infrared Sounder), SEVIRI (Spinning Enhanced Visible and InfraRed Imager), AMSU-A (Advanced Microwave Sounding Unit-A), MHS (Microwave Humidity Sounder), ATMS (Advanced Technology Microwave Sounder) and atmospheric motion vectors. Scatterometers provide information on ocean surface wind. Zenithal total delay signals and from radio-occultation measurements from the Global Navigation Satellite System (GNSS) are also assimilated.

With the advent of hyperspectral sounders such as AIRS and IASI, a variational bias correction (VarBC) method (Auligné et al., 2007) has been operationally implemented at Météo France and notably in the ARPEGE model. The VarBC scheme aims to minimize systematic innovations in radiances while preserving the differences between the background and other observations in the analysis system.

The observation operator allows to simulate observations from the model variables for comparison with the actual measurements. For satellites radiances, it includes a radiative transfer model. The accuracy of the forward model calculation could be limited by the accuracy of the NWP model, for some variables this is not sufficient to correctly model the observations and these observations have to be discarded.

The assimilation of clear radiances at Météo France is based on the McNally and Watts cloud detection scheme (McNally and Watts (2003)) The McNally and Watts (2003) scheme intends to detect clear channels and to assimilate channels unaffected by clouds even in a cloud-affected pixel. The channels are first re-ordered according to a ranking with respect to the altitude that reflects their relative sensitivity to the presence of cloud. After having applied a low-pass filter a search for the channel at which a monotonically growing departure can first be identified. Having found this channel all channels ranked more sensitive are flagged as cloudy and those ranked less sensitive are flagged clear.

In addition, a cloud characterization is made using cloud parameters (a cloud top pressure (CTOP) and an effective cloud fraction (Ne)) deduced from a $CO_2$-slicing algorithm ( Pangaud et al. (2009)). These two parameters are used to model the radiative impact of cloud as a single layer cloud, with an emissivity set to 1 using a clear sky radiative transfer model.

### 2.1.1 Main features of the IASI instrument

IASI is a key element of the Metop series payload of European polar orbiting meteorological satellites (Cayla, 2001). It was designed by CNES (Centre National d'Etudes Spatiales) in cooperation with EUMETSAT. The first flight model was launched in 2006 on board the first European meteorological satellite Metop-A in polar orbit. The second instrument, mounted on the Metop-B satellite, was launched in September 2012. The third instrument was mounted on the Metop-C satellite, which was launched in November 2018. The horizontal resolution of the instrument is 12 km at the nadir. IASI is dedicated to operational meteorological soundings with a high level of accuracy (specifications on temperature accuracy: 1 K for 1 km and 10% for humidity (Chalon et al., 2001)). Its measurements are also useful for atmospheric chemistry to estimate and monitor different trace gases such as ozone, methane or carbon monoxide on a global scale (Hilton et al., 2012).

IASI is a passive IR remote-sensing instrument using an accurately calibrated Fourier transform spectrometer to cover the spectral range from 3.62 $\mu$m (2760 $cm^{-1}$) up to 15.5 $\mu$m (645 $cm^{-1}$) with 8461 channels. Its spectral resolution is 0.5 $cm^{-1}$ with a spectral sampling of 0.25 $cm^{-1}$. The IASI spectrum can be divided into three major bands:

- from 645–1210 $cm^{-1}$ : $CO_2$, window and ozone channels mainly sensitive to temperature, called long-wave (LW) channels;

- from 1210–2040 $cm^{-1}$ : channels mainly sensitive to humidity, called water-vapour (WV) channels;

- from 2040–2700 $cm^{-1}$ : named short-wave (SW) channels.

Only a subset of 314 channels (300 channels selected by Collard (2007) and 14 additional channels for monitoring purposes) used in operations at Météo-France, is considered in this study

### 2.1.2 Towards the assimilation of cloudy infrared IASI radiances

Assimilation of cloudy radiances is a crucial challenge for NWP centres as the cloudy observations discard represent an under-exploitation of hyperspectral sounders especially in sensitive meteorological areas (McNally, 2002; Fourrié and Rabier, 2004).

As mentioned in the introduction, studies about all-sky infrared assimilation have started. The radiative transfer model RTTOV-CLD for cloudy sky, included in RTTOV version 11 (Saunders et al., 2013), offers a realistic modeling of the cloud scattering. This model also allows a better description of the cloud emissivity as well as cloud scattering, using the microphysical cloud profiles (water content, cloud ice content and cloud cover).

To simulate the radiances observed in cloudy conditions using RTTOV-CLD, we use two main types of clouds: firstly liquid water cloud which corresponds to two RTTOV-CLD cloud microphysical options depending on the land sea mask of the model (Stratus continental over land and Stratus Maritime over the sea) secondly the ice water cloud of the Cirrus type, using Baran parameterisation (Baran et al., 2014; Vidot et al., 2015) to define the optical properties.

To illustrate the benefit brought by RTTOV-CLD, Figure (1) shows IASI brightness temperature observations of a cloud-sensitive surface channel (1271, 962.5 $cm^{-1}$) and differences between the observations and the simulations computed with RTTOV considering clear-sky and with RTTOV-CLD. Brightness temperatures less than 250 K are usually associated with higher elevation cloud structures. By using RTTOV in clear sky (figure 1.b) to simulate IASI observations, the observation departures are mainly below zero and may reach up to -60 K. This can be explained by the fact that the main cloud structures associated with low values of brightness temperature for the surface channel are missing in the simulation. On the contrary, differences obtained with the RTTOC-CLD simulations are in overall in better agreement with lower positive and negative values (figure 1.c). No major differences are found for example for the cloud structures located over the North Atlantic (30N-70N, 40W-0W) and above (30S-70S, 60W-0W) the Southern Atlantic Ocean. Large difference values are mainly obtained in the Tropics region. This may be explained by the fact that clouds are better simulated in the ARPEGE model for mid-latitudes than in the Tropics.

## 3 Selection method of homogeneous observations

The assimilation of cloudy radiances in NWP models remains a ~~innovative~~ challenge. In the context of the preparation of all-sky assimilation, we plan to assimilate clear and cloudy observations that are completely covered in a homogeneous way, discarding the cases of fractional cloud observations.

These scenes are supposed to be better characterized and simulated than fractional cloudy scenes in NWP models. Indeed, by selecting homogeneous cloudy scenes in both model and observation spaces, we improve the agreement between observations and background simulations. This selection of cases seen as homogeneous by both IASI and the model avoids misplacement errors. In this section, limited to cases over sea to avoid problems related to the land surface properties, we describe several methods for analysing the homogeneity of the scene in the observation and model space. However these methods were applied over all surface in the assimilation experiments of section 4.

### 3.1 AVHRR clusters

In order to select homogeneous pixels, the AVHRR imager information collocated within IASI on the MetOp platform is used. The spatial resolution of AVHRR observations is around 1 km at nadir and measures the radiation emitted in six broad-band

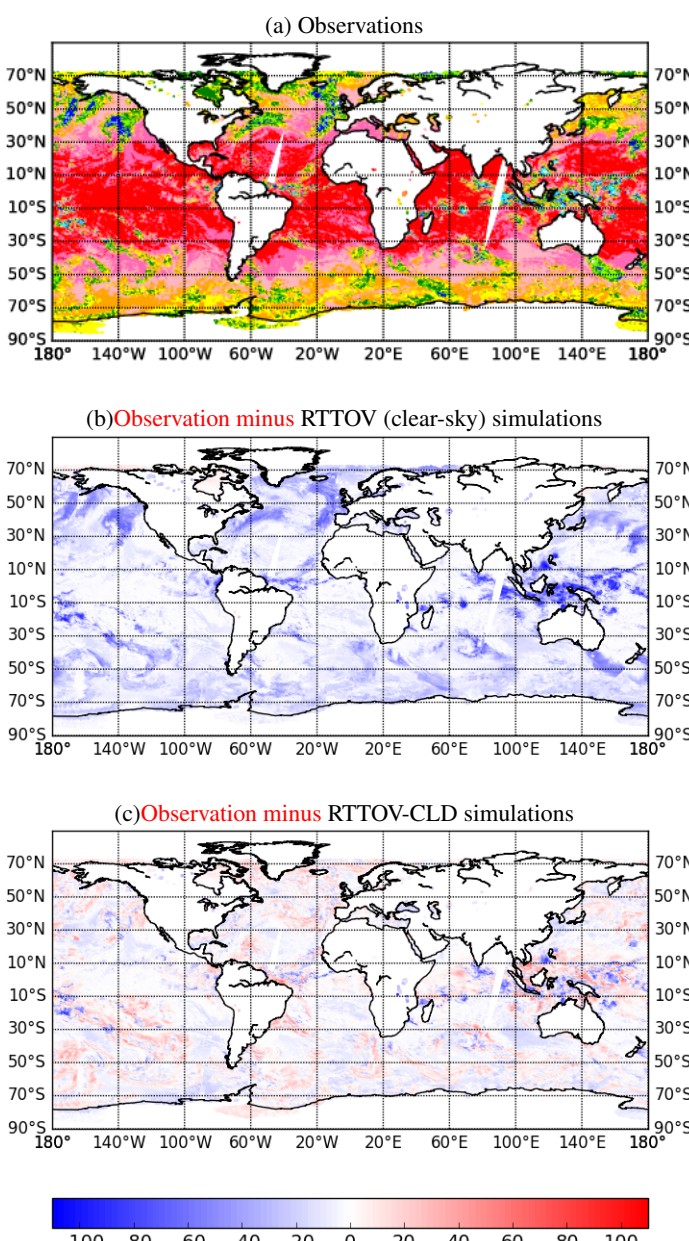

**Figure 1.** IASI brightness temperature (K) observations (a) from Metop A and B satellites and differences between observations and simulations using RTTOV (b) and RTTOV-CLD (c) for surface channel (1271, 962.5 cm$^{-1}$) for 30 January 2017 daytime over sea from ARPEGE 6-hour forecast fields.

channels: one visible channel, two near-infrared channels, a shortwave infrared channel and two long-wave infrared channels (10.5 $\mu$m and 11.5 $\mu$m). Two components of the IASI Level 1c ~~products~~ provided by EUMETSAT were used: the AVHRR clusters (Cayla, 2001) and the percentage of cloudy AVHRR pixels in the IASI FOV (product GEUMAvhrr1BCldFrac:Pequignot and Lonjou (2009)). The AVHRR pixels are clustered into homogeneous classes in the radiance space, (visible and infrared channels) using the K-mean classification algorithm. For each class and each AVHRR channel, the cluster product provides the coverage of the class within the IASI pixel, the mean and the standard deviation of AVHRR brightness temperatures within the class.

## 3.2   Selection criteria for homogeneous observations

This study intends to focus on those IASI pixels that contain only one cluster, which corresponds to a homogeneous scene. However only 2% of daytime IASI observations over sea contain only one class. The aggregation is built with all available AVHRR channels (visible, NIR, IR), several classes can be produced with the K-mean classification even with relatively small standard deviations for the IR channels. A IASI FOV with several classes, each one having a small standard deviation and a mean radiance close to the ones of the other classes, can thus be more homogeneous than a FOV with a single class but with very large value of standard deviations.

For this reason, the number of AVHRR clusters within each IASI pixel has not been used as a homogeneity criterion, but these characteristics have been used to calculate the overall AVHRR cluster statistics, aggregating the information provided by all clusters in the IASI FOV.

We tested four methods for selecting homogeneous scenes by calculating homogeneity criteria in the observation space as well as in the model space, using the AVHRR channels. The first two ones are described in the literature and we propose two other ones which are detailed below.

### 3.2.1   Homogeneity criteria derived from Martinet and al., (2013)

These homogeneity criteria are based on a single AVHRR infrared channel 11.5 $\mu$m, which is used to compute three homogeneity tests, the first two tests are calculated in the observation space and the third one in the model space:

**Intercluster homogeneity**

The intercluster homogeneity is based on $\sigma_{inter}$ defined as:

$$\sigma_{inter} = \sqrt{\frac{1}{\sum C_j} \sum_{j=1}^{N} C_j (L_j - L_{mean})^2} \tag{1}$$

Where $L_j$ is the mean radiances of cluster $j$ at channel 11.5 $\mu$m , $L_{mean}$ represents the radiance weighted average. The weighting is determined by $C_j$ is the cluster fraction of each class inside the IASI pixel. $N$ is the number of classes in the IASI pixel.

A small calculated standard deviation $\sigma_{inter}$ means that all classes observe a similar cloudy scene in the infrared channel. If this standard deviation is too high, each class observes a different scene (clear or cloudy) and the IASI pixel is very heterogeneous.

**Intracluster homogeneity**

In order to finalize the homogeneity criterion in the observation space, it is also necessary to check if each class itself is sufficiently homogeneous, using the following formula:

$$\sigma_{intra} = \sqrt{\frac{1}{\sum C_j} \sum_{j=1}^{N} C_j \sigma_j^2} \qquad (2)$$

Where the $\sigma_j$ are the standard deviations of each cluster $j$ calculated for the infrared channel 11.5 $\mu$m. The IASI observation is considered homogeneous if it verifies the following criteria:

  – Ratio between intracluster homogeneity $\sigma_{intra}$ and mean radiance $L_{mean} < 4\%$.

  – Ratio between intercluster homogeneity $\sigma_{inter}$ and mean radiance $L_{mean} < 8\%$.

**Background departure check**

Finally, in order to obtain a similar criterion in the model space, each AROME grid point within IASI FOV was used to simulate the equivalent AVHRR channel 11.5 $\mu$m with RTTOV-CLD. Homogeneous IASI observations are preserved if the ratio of the standard deviation of the AVHRR simulations and the simulated mean radiance of the AVHRR is less than 8%.

**Adaptation of the method**

In the original Martinet et al. (2013)) study, the third check verifying that both the observation and the model observe the same cloudy scene was done with the difference between the mean AVHRR brightness temperatures from the observed and simulated clusters less than 7 K. Here, the ARPEGE model has a coarser resolution and it is not possible to simulate the AVHRR clusters. This check was adapted with the difference between the AVHRR observation and the AVHRR simulation from the guess, which come from an horizontal interpolation of the 12 profiles surrounding the observation position coming from a 6-hour forecast.

~~In our study, which focuses on the ARPEGE global model, we chose to use the simulated brightness temperature from the guess profiles coming from a 6-hour forecast and interpolated using 12 points surrounding the observation position.~~ The homogeneous cases are retained as long as the difference between AVHRR observations and simulations is less than 7 K. This method will be noted M2013.

### 3.2.2 Homogeneity criteria derived from Eresmaa (2014)

The study of Eresmaa (2014) proposed an imager assisted cloud detection for the global ECMWF NWP system and was based on the hypothesis that each AVHRR cluster are made of fully clear or fully cloudy pixels.

Therefore, his selection criteria is only intended to diagnose and retain observations when they were completely clear, using the last two infrared channels of AVHRR (10.5 $\mu$m and 11.5 $\mu$m). This detection is based on three checks called the homogeneity check, the intercluster consistency check and the background departure check. If a IASI pixel do not satisfy one of these checks, it is not free of cloud and is rejected.

The standard deviation of the brightness temperatures of the two infrared channels from all pixels present in the FOV is used for the first check. If both standard deviations are over the pre-determined threshold values (0.75 and 0.80 K, respectively), it means that a cloud is potentially observed and the IASI observation is rejected. The intercluster consistency check relies on the comparison between the properties of the different clusters within the IASI FOV. The distance of each cluster to the background in both infrared AVHRR channels as well as the distance between each pair of clusters. A cloud is detected if there

is a pair of clusters covering more than 3% of the IASI FOV and for which the intercluster distance exceeds the minimum value of the distances between these clusters and the background.

The distance between 2 clusters $j$ and $k$ is computed as the squared-summed intercluster departure:

$$D^{jk} = \sum_{i=4}^{5} (R_i^j - R_i^k)^2 \tag{3}$$

where $R_i^j$ is the mean brightness temperature of cluster $j$ for channel $i$. In addition the distance of the cluster $j$ to the

background is computed with:

$$D^j = \sum_{i=4}^{5} (R_i^j - R_i^{BG})^2 \tag{4}$$

Where $R_i^{BG}$ is the clear-sky background brightness temperature for AVHRR channel $i$. The observation is rejected due to the diagnostics of the presence of a cloud if the following inequality is true and the coverage of clusters $j$ and $k$ is over 3%:

$$D^{jk} > min(D^j, D^k) \tag{5}$$

The last check on the background departure is computed as a fractional-weighted mean of the squared-summed background departures:

$$D_{mean} = \sum_{j=1}^{N} D^j C_j \tag{6}$$

where $N$ is the number of clusters in the IASI FOV and $C_j$ is the fractional coverage of cluster $j$. The presence of cloud is diagnosed if $D_{mean}$ exceeds the threshold value of 1K².

**Adaptation of the method**

Since this method assumes that each cluster is made of pixels that are either all clear or cloudy, its homogeneity tests have been adapted to the selection of clear and cloudy pixels, with criteria that would fit our purpose, with the first test in the observation

space and the second one background departure check. All AVHRR simulations from background are made with RTTOV-CLD and the threshold of the background departure check was modified.

~~-Inter-cluster homogeneity.~~

~~This test uses the standard deviation of the infrared brightness temperature, calculated on all clusters occupying the IASI field of view. The standard deviation is calculated in the same way as Eresmaa (2014) and the IASI pixel is considered homogeneous if the two standard deviations (one for each channel) are below their predetermined threshold values of 0.75 K and 0.8 K respectively.~~

~~-Background departure check. In this test,~~ We used the $D_{mean}$ proposed by Eresmaa (2014) ~~but the IASI pixel is considered as homogeneous~~ to perform here a kind of cloudiness consistency check between the observation and the model simulation if $D_{mean}$ is less than 49 K$^2$. This particular value of threshold allows to keep more than 50% of the observations compared to the initial threshold of 1K$^2$ by Eresmaa (2014) which retains only 18% of the observations. In addition this threshold compares well with the one applied by M2013, but it is applied over the 2 IR AVHRR channels. This method is referenced as E2014 in the following.

The threshold values of the homogeneity criteria derived from Martinet et al. (2013) and Eresmaa (2014), are based on the analysis of statistics, applied to all IASI FOVs of the different situations (day/night at sea). Threshold values are specified in such a way that the standard deviation between the observations and simulations is not too large while keeping a fair amount of the observations.

### 3.2.3   Selecting homogeneous scenes in observation space

This third method, (called Obs_HOM thereafter) proposes a homogeneity check in the brightness temperature space calculated only on the observation space, using both infrared AVHRR channels (10.5 $\mu$m and 11.5 $\mu$m). It is the same test as in M2013 but in the brightness temperature space. This inter-cluster homogeneity criterion is based on the relative standard deviation of AVHRR clusters inside the IASI pixel. This test is satisfied when all classes observe a very similar scene in the AVHRR infrared channels. To evaluate the interclass homogeneity, the standard deviation of the mean brightness temperature of clusters which occupy the IASI FOV has been calculated using the following formula:

$$\sigma_{inter} = \sqrt{\frac{1}{\sum C_j} \sum_{j=1}^{N} C_j (R_{i,j} - R_{i,mean})^2} \tag{7}$$

Where: $R_{i,j}$ is the mean brightness temperature of cluster $j$ on channel $i$, $R_{i,mean}$ represents the weighted average on channel $i$, $N$ is the number of classes in the IASI pixel, and $C_j$ is the cluster fraction.

Figure 2 provides a calibration to determine the thresholds to be used to define homogeneous scenes. These thresholds should lead to a sufficient size of the selected dataset and avoid selecting the fractional cloud as much as possible. Therefore we decided to select an observation if the ratio between intercluster homogeneity and mean radiance for both AVHRR IR channels (10.5 $\mu$m and 11.5 $\mu$m) are less than 0.8%. If this test is only applied over channel (10.5 $\mu$m), 68,2% of the observations are

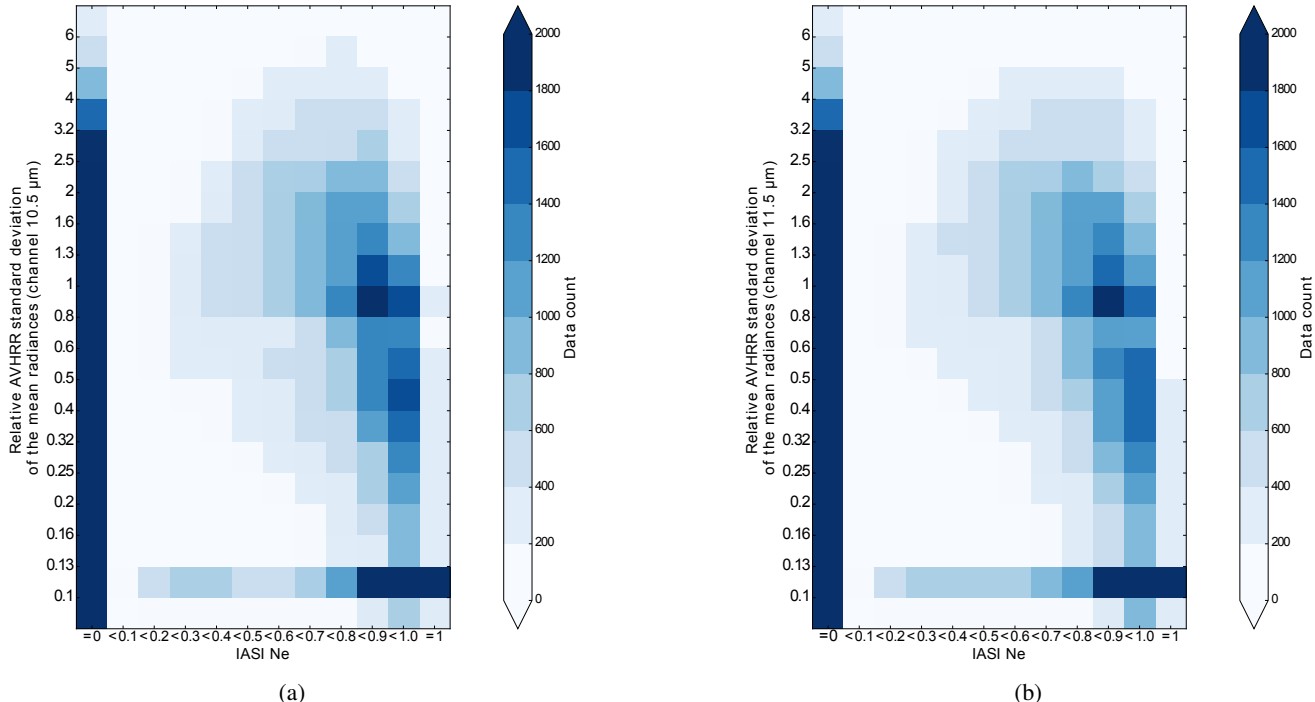

**Figure 2.** Density plot (number of observations) of the values of effective cloud fraction retrieved from IASI by a $CO_2$-slicing algorithm (on the abscissa) with respect to the relative cluster standard deviation of the mean radiances (%, on the y-axis) for intercluster homogeneity for (a) the AVHRR IR channel (10.5 $\mu$m) and (b) the AVHRR IR channel (11.5 $\mu$m)

selected, if applied over channel (11.5 $\mu$m), 69,6% of the data are kept and if it is applied over both channels, 67,3% pass the test.

### 3.2.4 Compromise for the homogeneous scene selection

Based on the previous methods, we propose a fourth one which represents a compromise between them. Two AVHRR infrared
5  channels (10.5 $\mu$m and 11.5 $\mu$m) are used, and we define two homogeneity criteria in the observed and simulated brightness temperature spaces.

The first criterion for homogeneity is the interclass homogeneity check which was used in the third method, calculated in the observation space (presented in section 3.2.3). Similarly, we used the background departure check in the observation space $D_{mean}$ (presented in section 3.2.2).
10  Only observations that fulfilled the two following criteria were selected:

- Ratio between intercluster homogeneity and mean brightness temperature for two AVHRR IR channels (10.5 $\mu$m and 11.5 $\mu$m) $< 0.8\%$.

- Sum of the average distances between each cluster and the background $< 49$ K².

| Methods | Literature | AVHRR channels used | Homogeneity criteria in observation space | Test on background simulation |
|---------|-----------|---------------------|-------------------------------------------|-------------------------------|
| M2013 | Martinet et al. (2013) | 11.5$\mu$m | intra and intercluster | distance with observation |
| E2014 | Eresmaa (2014) | 10.8 and 11.5$\mu$m | intercluster | average distance with each cluster |
| Obs_HOM | | 10.8 and 11.5$\mu$m | intercluster | No |
| COMPR | | 10.8 and 11.5$\mu$m | intercluster | average distance with each cluster |

**Table 1.** Summary of the criteria for homogeneous IASI observation selection used in this study.

This method is named COMPR in the following. All the four methods are sumerized in Table 1.

## 4 Inter-comparison of selection criteria

We applied our selection criteria on January 30, 2017 and result from an observation sample composed of 188090 IASI FOV during the daytime over the sea are presented. Same conclusions were found for the other cases (night-time and/or over land).

| | Number of Observations | Homogeneous Clear Observations | Homogeneous Cloudy Observations | Heterogeneous Observations |
|---|---|---|---|---|
| All observations | 67599 | 12% | 51% | 37% |
| M2013 | 57% | 11% | 27% | 20% |
| E2014 | 26% | 10% | 9% | 7% |
| Obs_HOM | 69% | 12% | 35% | 22% |
| COMPR | 40% | 10% | 17% | 13% |

**Table 2.** Evaluation of heterogeneity inside the IASI pixel with respect to the SEVIRI cloud type.

5   An evaluation of the homogeneity inside the IASI pixels according the various selection criteria was performed with independent SEVIRI data over the sea to check that the retained observations are selected for clear or overcast scenes. Here the considered data were the cloud type from the geostationnary SEVIRI sounder which is a product from the Satellite Application Facility on support to NoWCasting and very-short range forecasting of EUMETSAT. For each IASI pixel, the cloud types of the 4 SEVIRI pixels closest to the IASI centre were compared. If the 4 cloud types were sea, the IASI pixel is considered as clear,

10 if the same cloud type is found for SEVIRI pixels, the IASI observation is set to homogeneous cloudy observation, otherwise it is considered as a heterogeneous observation. Table 2 presents the results of this comparison. 67599 IASI observations were colocated with SEVIRI data during the day of 30 January 2017. In the global dataset, we found that 12% are clear homogeneous observations, 51% are cloudy homogeneous observations and 37% of IASI observations are made of different SEVIRI cloud type. This corresponds well to the results obtained with the results found with the percentage of cloudy AVHRR pixels

15 in the IASI field of view (Table 3). When applying the M2013 criteria, 11% of clear observations are retained. this number

| | Number of observations | Cloudy observations CldCover=100 | Clear observations CldCover=0 | Partially cloudy observations 0<CldCover<100 |
|---|---|---|---|---|
| All observations | 188090 | 50% | 12% | 38% |
| M2013 | 54% | 19% | 10% | 25% |
| E2014 | 22% | 6% | 10% | 6% |
| Obs_HOM | 67% | 32% | 13% | 22% |
| COMPR | 36% | 11% | 10% | 15% |

**Table 3.** Overview table of statistics obtained with the different homogeneity criteria : number of observations retained, percentage of cloudy observations (cloudcover of 100), percentage of fully clear observations (cloudcover of 0). Percentage are given with respect to the whole number of observations.

does not vary a lot when applying the other criteria. The number of homogeneous cloudy observation is more sensitive to the criteria because it varies between 9% for E2014 and 35% for the Obs_HOM one. The COMPR criteria provide a compromise between M2013 ad E2014 in terms of selected observations.

The percentage of cloudy AVHRR pixels in the IASI field was also used to assess the choice of homogeneity criteria (Table 5   3).

Our global dataset is made of 50% of the observations entirely covered by clouds and 12% of clear observations according to the AVHRR cloud cover. These results obtained over the globe set agree well with the ones obtained with SEVIRI data over the Atlantic Ocean. The percentage of selected observations for each selection method is larger (+2/4 %) with the SEVIRI data evaluation than with the AVHRR cloud cover.

10   The bias and standard deviation of observations minus simulations (O-G), are shown in Figure 3.(a) for the 314 IASI channels. As expected, the best statistics are obtained for channels less affected by clouds (e. g. CO2 and water vapour high peaking channels).

For the whole dataset, window channels present a bias of around -0.6K. The standard deviations are larger (around 12 K) for window channels sensitive to the surface. With the M2013 selection method (figure 3. (b)), the standard deviation of 15   window channels is reduced to around 4 K as well as the bias close to zero. The standard deviation of the other channels (680-780cm$^{-1}$) is also well decreased. The E2014 selection method (figure 3. (c)) improves the bias and the standard deviation (2.0 K for window channels) for all the channels. As expected, the impact is larger for surface sensitive (and thus cloud sensitive) channels than for the tropospheric channels (680-780cm$^{-1}$). On the contrary, with the Obs_HOM method (figure 3. (d)), small statistics improvement is obtained for the standard deviation and the bias. The statistics obtained with the COMPR 20   method (Fig. 3.(e)) are reduced compared to the whole dataset and slightly less good than with the initial E2014 method (for window channels the standard deviation is around 2.2 K instead of 2 K for E2014).

To complete the comparison, the Probability Density Function (PDF) of the O-G differences was studied (Fig.4). Three channels were assessed: the window channel 1271 (962.5 $cm^{-1}$, whose weighting function peaks at around 1000 hPa), the

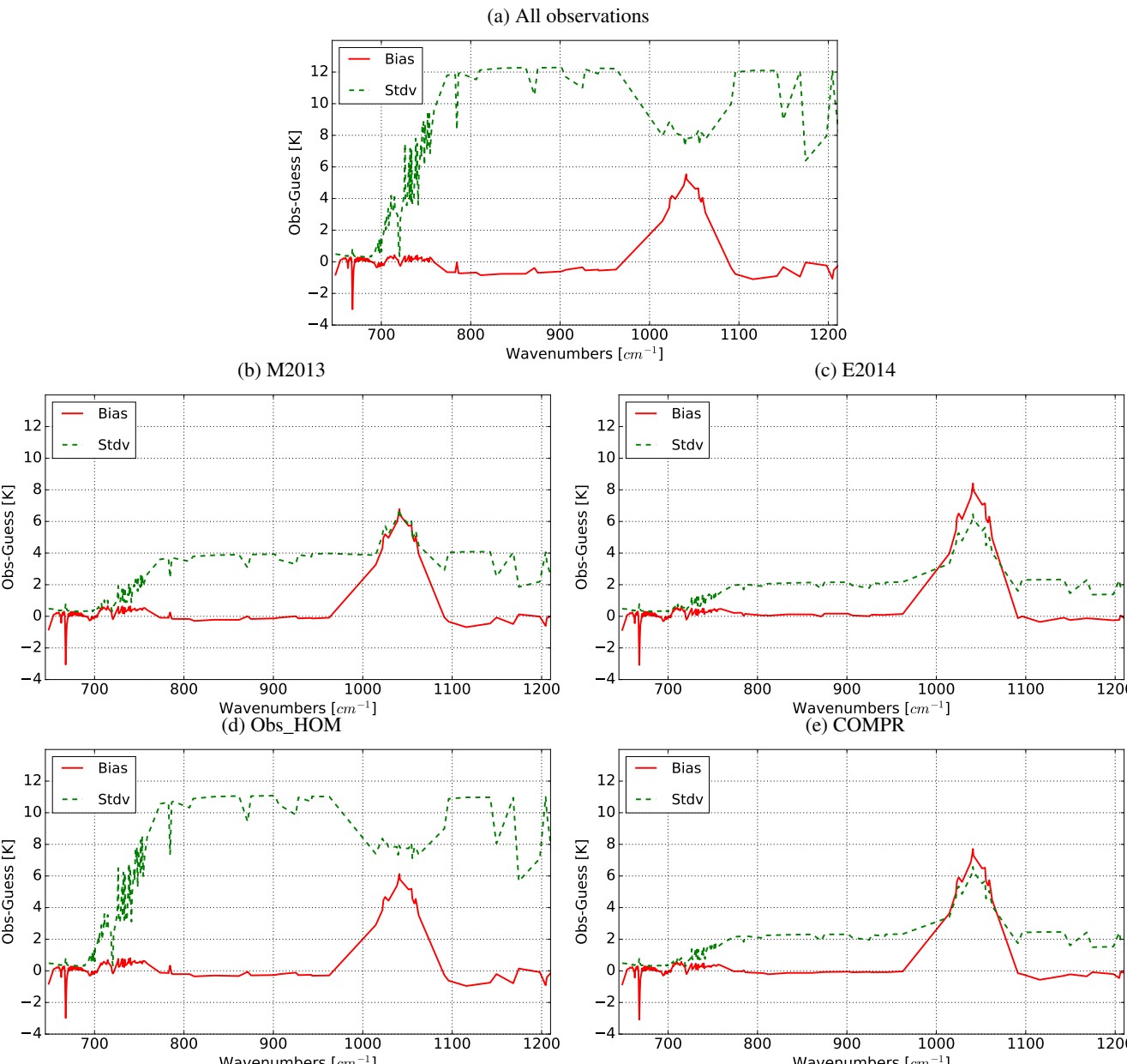

**Figure 3.** Bias (red solid line) and standard deviation (green dashed line) in Kelvin (K) of the differences between IASI observations and background simulations using RTTOV-CLD and a 6-hour forecast: (a) for the whole dataset, (b) after applying the homogeneity criteria derived from (Martinet et al., 2013), (c) after applying the homogeneity criteria derived from (Eresmaa, 2014), (d) after applying the selecting homogeneous scenes based on observation space, (e) after applying the compromise to select the homogeneous scenes. Observations are for January 30, 2017.

mid-tropospheric water vapour channel 2701 (1320 $cm^{-1}$, weighting function maximum at around 400 hPa) and the low-tropospheric water vapour channel 5403 (1955 $cm^{-1}$, weighting function peaking at around 900 hPa),

~~The distribution asymmetry is relatively small for mid and low tropospheric water vapour channels.~~ The distribution asymmetry is reduced for mid and low tropospheric water vapour channels with M2013 and E2014 selection. The impact of clouds

is evident on the window channel, with differences ranging from -90 to 64 K. After the homogeneity criteria is been applied, narrower Gaussian distributions are observed for all channels with a significant improvement for the window channel. Using the M2013 criteria, differences in O-G for the window channel are significantly reduced, from -18 K to 20 K, and from -7 to 9 K using the E2014 criteria (Figure 4.g, h, i).

With Obs_HOM criteria (Figure 4.(j, k, l)), the O-G distribution is not much improved for all channels ~~except for the~~

~~low-tropospheric water vapour channel where the range is reduced from 60 K to 40 K.~~ When the homogeneity criterion in the model space is added using the COMPR selection, the O-G distributions become symmetrical , get closer to the gaussian distribution ~~(and Gaussian)~~ and centrered around zero for the three previously selected channels(Figure 4.(m, n, o)), which indicates the data are correctly diagnosed as homogeneous.

| | Number of observations | Bias temperature channels (650-770) $cm^{-1}$ | Stdev temperature channels (650-770) $cm^{-1}$ | Bias (Window channels) (770-980) $cm^{-1}$ | Stdev (Window channels) (770-980) $cm^{-1}$ |
|---|---|---|---|---|---|
| All observations | 188090 | 0.06 K | 2.53 K | -0.60 K | 11.7 K |
| M2013 | 54% | 0.14 K | 0.82 K | -0.16 K | 3.7 K |
| E2014 | 22% | 0.13 K | 0.59 K | 0.11 K | 2.0 K |
| Obs_HOM | 67% | 0.19 K | 2.22 K | -0.20 K | 10.5 K |
| COMPR | 36% | 0.12 K | 0.64 K | -0.09 K | 2.1 K |

**Table 4.** Overview table of statistics obtained with the different homogeneity criteria : number of observations retained, bias and standard deviation computed for the channels included in the range between 650-770$cm^{-1}$ and 770-980 $cm^{-1}$.

Table 4 summarizes statistics in terms of bias and standard deviation of background departure for the different datasets. The

bias and standard deviation obtained by the M2013 method have some reasonable statistics before the assimilation (-0.6 K for the bias and 3.7 K for the standard deviation, for the window channels). The E2014 selection method seems relevant for selecting homogeneous scenes in terms of bias and standard deviation (0.11 K and 2.0 K respectively, for the window channels). However, the number of selected observations presents a disadvantage for this selection method, since E2014 method keeps only 22% of the observations of which 10% are totally clear, 6% are totally covered by clouds and 6% are heterogeneous.

These observations are distributed throughout the globe, but we keep more observations on high latitudes.

The Obs_HOM method allows to keep 67% of observations, which 12% are totally clear and 32% are totally covered by clouds, but this method does not give acceptable statistics (bias of -0.2 K and standard deviation of 10.5K). When the test on observations minus simulations of the infrared channels AVHRR are added by the COMPR method, results are improved. For window channels the bias is reduced to -0.09 K and the standard deviation to 2.1 K compared to -0.6 K and 11.7 K for all obser-

vations, which presents a good score compared to the M2013 and Obs_HOM methods. In addition 36% of the observations are retained, compared to the whole dataset, with 10% of clear observations and 11% of cloudy observations of the total amount, which represents twice the amount of cloudy observations selected by E2014, which removes many more observations, and shows that the proposed methodology is effective.

The cloud cover distribution corresponding to the amount of observations that is kept (36%) is made of 28% of clear observations and 29% of the observations totally covered by clouds. In addition, 14% of the observations have a cloud cover of less than 10% and 4% of the observations have a cloud cover exceeding 90%. The observations kept are distributed in different parts of the globe (Figure 5.a) although we have been able to retain different cloud types, including high clouds even in the tropics for few cases only (Figure 5.b). This may be explained by the weakness of the model clouds in these areas.

The main objective of the study is to select homogeneous IASI observations in clear and cloudy sky which are well simulated with RTTOV-CLD and could be used in data assimilation. Comparison of different methods of selecting homogeneous scenes showed that the M2013 method improves the first guess departure statistics (bias of -0.16 K and standard deviation of 3.17 K) but it keeps more heterogeneous observations (25%) according AVHRR cloud cover than the E2014 method, which significantly improves the statistics (bias of 0.11 K and standard deviation of 2 K) and favours more clear observations but

keeps only 22% of the observations. The Obs_HOM method, which focuses only on homogeneity in the observation space, does not strongly improve the statistics but it filters 33% of heterogeneous observations. However the addition of the criterion on the simulated observations in the COMPR method improves the scores on IASI simulations (bias of 0.09 K and standard deviation of 2 K), while retaining 36% of the observations and among them a similar part of homogeneous clear and homogeneous cloudy observations. This data selection representing a compromise between M2013 and E2014 is chosen for a sata

assimilation experiment.

## 5   Impact on NWP analyses and forecasts

After the selection criteria were implemented in the assimilation system of Météo France, their impact was tested through a 4D-Var assimilation experiments in the ARPEGE global model. The impact of the homogeneity criteria for data selection on all observation simulations, on analyses and forecasts is evaluated.

### 5.1   Experimental design

To evaluate the impact of our homogeneity criteria on the assimilation process over sea and land, during daytime and night, ~~four~~ experiments were performed over one month from 06/12/2017 to 17/01/2018. 314 IASI channels were used in the simulation, and 129 channels ~~(Tables ?? and ?? in Appendix)~~ were used for assimilation as operationally.

The first experiment is the reference (REF), where IASI observations are assimilated with all other observation type as in

the operational system at Météo-France.

~~The Obs_HOM criteria for selecting homogeneous IASI observations (presented in the 3.2.3 section) are applied in the second experiment called (EXP.A) on top of the Mc Nally and Watts cloud detection. Finally in the third~~ In a second experiment

called (EXP), we applied our COMPR approach (presented in the 3.2.4) on top of the Mc Nally and Watts cloud detection. These ~~sets of~~ experiments aim to evaluate the impact of the COMPR method of selecting homogeneous IASI observations on simulation and assimilation processes. ~~The following Table ?? details the main features of each experiment.~~

In these experiments, no cloudy observations detected with the CO2-slicing method and used with a single grey-cloud layer scheme was assimilated unlike Guidard et al. (2011). We focus on clear sky assimilation. RTTOV-CLD was only used to compute the homogeneity criteria based on cloudy AHVRR simulations and RTTOV was used for the clear sky assimilation.

## 5.2 Impact on observation

~~The number of assimilated channels for each observation is different depending the areas, e.g. in the tropics, there are less clear channels (between 30 and 40 channels) in the REF (Figure ??.a), EXP.A (Figure ??.b) and EXP.B (Figure ??.c), which is explained by the presence of high clouds in this region. In EXP.A(Figure ??.b) with the Obs_HOM criteria, some observations were filtered in the tropical area, and even more in EXP.B(Figure ??.c) where the criterion used is even more stringent, more observations are filtered in areas corresponding to high clouds.~~

Figure 6.a gives the number of assimilated observations into the ARPEGE model as function of the IASI wavenumber for REF ~~, EXP.A and EXP.B~~ experiment. The order of magnitude is below $8.10^6$ whichever the wavenumber considered and for both experiments showing that among the 129 IASI channels selected for the assimilation, the occurrence proportion are well balanced. However, the assimilated number changes between experiments depending upon the spectral band considered. Four spectral band can be mentioned:

- $[657, 687.25]$ $cm^{-1}$ wavenumbers range corresponding to the stratospheric temperature channels keeps the same number of observations in both experiments because these channels are not affected by the presence of clouds.

- $[726.5, 1421]$ $cm^{-1}$ wavenumber range, corresponding to tropospheric and surface temperature channels and also to mid and high tropospheric water vapour channels, the number of observations is decreased by 15% for EXP (Figure 6.b).

- $[1800, 2015.5]$ $cm^{-1}$, finally the number of low tropospheric water vapour sensitive channels is slightly decreased ~~for EXP.A by 0.5% and~~ between 8% and 14% for EXP (figure 6.b).

## 5.3 Impact on background and analyses

The analysis departure data discussed below are made by comparing the analysis between the REF and the EXP experiments ~~(EXP.A and EXP.B)~~ to evaluate the impact of the criteria for selecting homogeneous IASI observations ~~(refer to Table ??)~~.

Figures 7.a and 7.b present the impact of COMPR criteria in the temperature and humidity analyses of the first assimilation cycle. This implementation removes some IASI observations from the assimilation and this reduction has an impact on the analysis. In Figure 7.a, a negative temperature difference is located in the Atlantic Ocean near to the South-West African coast. ~~A similar behaviour on temperature is noticed from the impact of COMPR (EXP.B) as shown in Figure 7.c.~~ Weaker and patchy impact is reported on specific humidity that is mainly located in the tropics. EXP seems to remove some temperature and humidity analysis increments from the REF experiment just at some isolated locations.

In order to assess the impact of the new selection of IASI observations on the analyses and forecast, first guess departures (FG departures) corresponding to the difference between the observations and the simulation from the 6-h forecast and the analysis departures (AN departures) are computed. As biases and standard deviations of FG and AN departures were very weak for IASI, CRIS and AMSU-A instruments, and humidity measurements performed by radiosondes, relative differences have been performed between experiments to highlight detailed comparisons (Figure 8). Negative differences are related to a reduction of the standard deviation with respect to REF one and thus to an improvement.

For IASI (Figure 8.a), regarding exclusively the significant differences with a 95% value, FG departure standard deviation was mainly reduced or unchanged in EXP compared to REF depending upon the wavenumber; increases are observed around 2000 $cm^{-1}$ and 1400 $cm^{-1}$ while significant reduction of 0.5-1 % is seen between 1000 and 1320 $cm^{-1}$. The standard deviation reduction on AN departures can be noted at around 2000 $cm^{-1}$ and between 950 and 1320 $cm^{-1}$.

Concerning the CrIS observations (Fig. 8.b), the differences results are mainly not significant excepted at around the 850 water vapour channels where the standard deviation increases for both FG and AN departures, and a significant improvement for the AN departures at the channels 160.

Results obtained for AMSU-A (Fig. 8.c) are mainly satisfactory with FG departure standard deviation differences reduced by around 0.05 % for channels 6, 9, 10, 11 and 12 ~~with the EXP.A and EXP.B~~. However, a significant degradation of 0.2 % is observed for channel 8. The AN departure results follow more or less the same behaviour. Finally, no significant standard deviation difference is observed concerning the TEMP-q observations (Figure 8.d).

Results shown in this part report small but non-negligible impact of the homogeneous criteria implemented into EXP on the analyses and the short range forecasts. Indeed, as seen in the previous section (section 5.2), selected IASI observations are removed over the more cloudy locations and then impact the humidity and temperature analyses (as seen in Figure 7). Statistical results in Figure 8 report a non-negligible decrease of the dispersion within the FG departure and AN departure for IASI observation and AMSU-A for several channels but some negative impacts have to be noted for other wavenumbers. More attenuated and mainly non-significant impacts can be recorded for CrIS and TEMP-q observations. Thus, the analyses and short range forecasts have been slightly changed compared to REF.

## 5.4 Impact on forecast scores

The forecasts from EXP at 00UTC for the period 7 December 2017 to 17 January 2018 were compared to REF ones and evaluated against radiosondes and operational analyses from ECMWF. Rootmean square forecast errors at the 12-h forecast ranges with respect to the ECMWF analyses were computed for temperature, relative humidity and wind. Similar computations were made against radiosondes . No major difference can be found between both experiments. Very small improvements of the 12-hour forecast with respect to the ECMWF analyses were found in the Southern hemisphere for temperature and wind at around 700 hPa (Figs. 9). This reduction of 2% for temperature and 0.5% for the wind is significant accordin a Bootstrap test with a 99.5% confidence level. Other improvements are found at 200 hPa for temperature (1.5%) and at 500 hPa for wind (0.5%). Regarding the evaluation against radiosondes, very small, but not significant, improvements for the wind were found in the troposphere in the Southern hemisphere and in the Tropics.

## 6   Conclusion and perspectives

A new method using of collocated AVHRR cluster information to improve the selection of homogeneous IASI observation scenes within the numerical weather prediction ARPEGE model has been developed at Météo-France for data assimilation purposes and has been presented in this study.

The first step consisted in adapting the IASI observation operator based on the RTTOV radiative transfer model by using the RTTOV-CLD module with cloudy microphysical parameters (liquid water content (ql), ice content (qi) and cloud fraction) for the simulation of cloudy radiances. A qualitative evaluation of such module showed realistic simulated cloud structures at various locations around the globe with a quite good agreement against IASI observations.

The second and main step of this work was to assess the impact of several methods used to select homogeneous IASI observations using AVHRR clusters. Two selection methods (derived from the literature : Martinet et al. (2013) and Eresmaa (2014))) were preliminarily evaluated. Despite a good improvement in terms of biases and standard deviations of the FG departures, ~~it was found that these two methods were not satisfactory in an operational context (in assimilation) to a large IASI observation reduction.~~ the criteria from the Martinet et al. (2013)'s method favours the homogeneous cloudy observations and retains more than a half of the observations while the Eresmaa (2014)'s method gives priority to clear observations and keeps only 22% of the observations. Then, two ~~new sets of~~ criteria were defined from these two previous methods in order to have a more balanced choice of clear and cloudy observations and good statistics in terms of bakcground departures and implemented within the ARPEGE model :

- The first criterion derived from Martinet et al. (2013) method  looks for the consistency between different clusters occupying the same IASI FOV by examining this homogeneity relative to the weighted average brightness temperature of the AVHRR clusters; it is only based on observations and computed for both infrared AVHRR channels as in Eresmaa (2014). This criterion allows to retain 67% of observations.

- In addition, the second criterion is derived from Eresmaa (2014)'s test and assesses the coherence of each cluster compared to the background brightness temperature simulation; it is in fact a good compromise between ~~the previous criterion and~~ the two "historical" ones with accurate statistics and a sufficient number of observations 36% that passed the check. It also allows to retain the same proportion of homogeneous clear and cloudy observations contrary to the derived Martinet et al. (2013) and Eresmaa (2014) methods.

Therefore, assimilation experiments were conducted to assess the impact of these new selecting homogeneous IASI observation features in the current clear sky assimilation. This revised check was added to the McNally and Watts (2003) cloud detection. The results obtained in this case show that the scenes categorization has been facilitated and cloudy observations can be better filtered out compared to what is done in the operational ARPEGE version. 3% of all observations are rejected with the compromise method ~~and only 1% for the method based only on homogeneity in observation space which is more convenient for the assimilation~~. The impacts on the first guess and analysis departures (showing more Gaussian shape) are generally low but with a beneficial reduction on the standard deviation of first guess departures mainly on the IASI and AMSU-A observa-

tions. Regarding the forecasts scores, neutral impact is reported when these selection criteria are taken into account on top of the McNally and Watts (2003) algorithm.

However, this step has been necessary to prepare the future which will consist of the assimilation all sky within the ARPEGE model. This method of observation selection allows to separate the clear-sky and cloudy scenes and manage each route in an independent way. Then, it could be available to directly assimilate the cloudy radiances into the 4D-Var ARPEGE by adapting the observation errors for more cloudy situations. However, hydrometeors used in the RTTOV-CLD are not available into the background error covariance matrix and then cloudy and convective situations are badly represented and will penalise the cloudy direct assimilation. In order to bypass this problem a the second solution under study is to retrieve information within cloudy observation by a Bayesian inversion method, in a first step, and assimilate these retrieved products in terms of temperature and/or humidity profiles into the 4D-Var in a second step. This method called 1D-Bayesian + 4D-Var was already studied for microwaves radiances (Guerbette et al., 2016; Duruisseau et al., 2018) and is successfully used since 2010 for radar reflectivities (Wattrelot et al., 2014) assimilation within the AROME convective scale model.

*Competing interests.* No competing interests are present here.

*Acknowledgements.* CNES and Météo-France funded this work through a PhD grant for Imane Farouk. Cloud classification data from MSG are provided by Météo-FranceCMS. We thank the ICARE Data and Services Center for providing access to the data used in this study. Jean-Francois Mahfouf and Jean Maziejewski are warmly thanked for their help in revising a previous version of the manuscript.

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

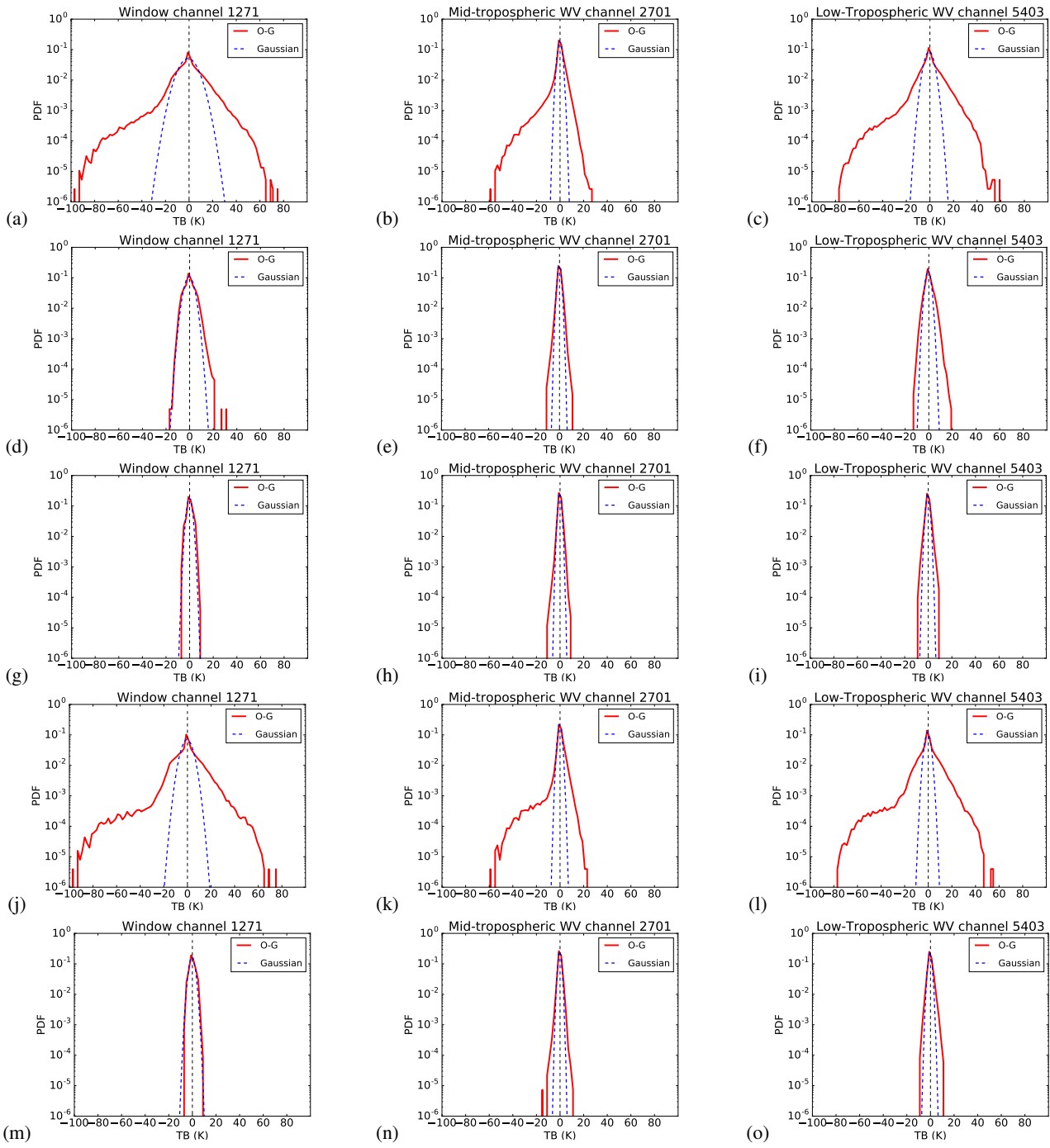

**Figure 4.** Frequency distribution of brightness temperature difference between observation and background (O-G) for all observations (a, b, c), after applying the homogeneity criteria derived from Martinet et al 2013 (d, e, f), the homogeneity criteria derived from Eresmaa 2014 (g, h, i,), the third method based on observation space method (j, k, l) and the compromised approach (m, n, o). The PDF are presented for three channels: window channel 1271, low-tropospheric water vapour channel 5403, and mid-tropospheric water vapour channel 2701). The Gaussian distributions with the same error characteristics (mean and standard deviation) are also shown in blue dashed lines.

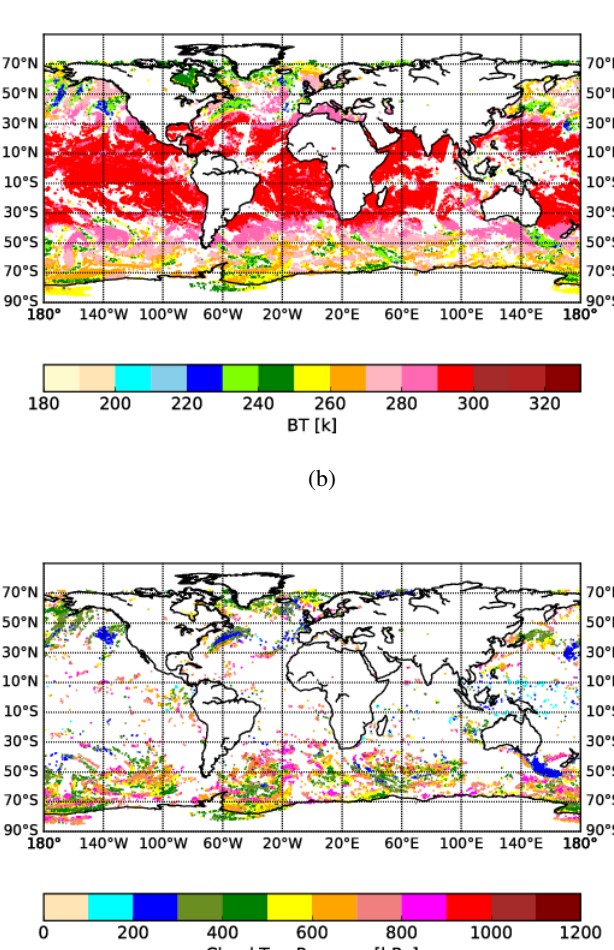

(a) COMPR

(b)

**Figure 5.** Map of IASI observations of brightness temperature (K) for surface channel (1271, 962.5 $cm^{-1}$), after applying the COMPR method (a), cloud-top pressure (hPa) observations retrieved from a $CO_2$-slicing algorithm applied on IASI data (b), for 30 January 2017 daytime over sea.

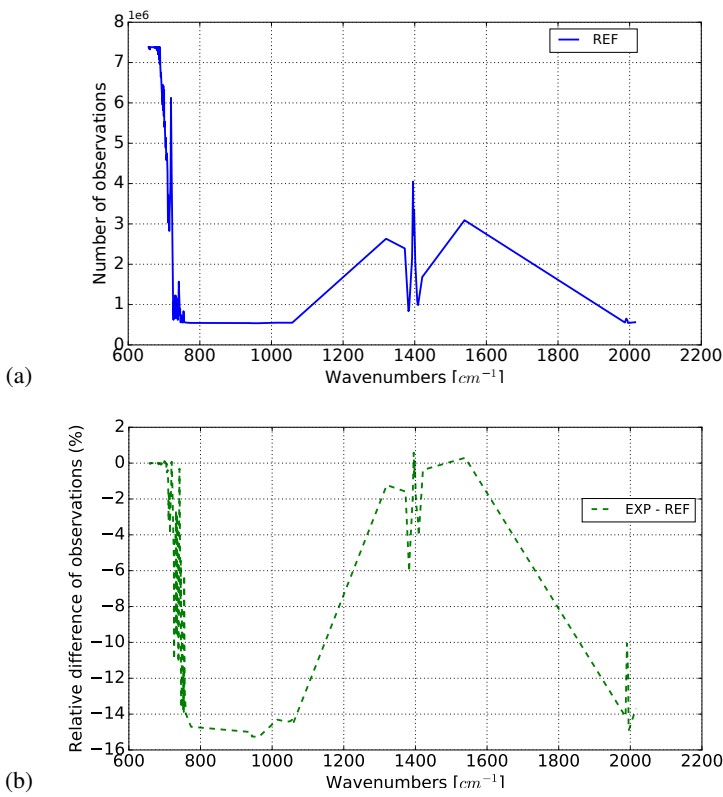

(a)

(b)

**Figure 6.** Number of assimilated IASI data over the whole experimental period (41 days) as a function of wavenumber of IASI for the REF experiment (a) and relative difference of number of assimilated observations (b) for the EXP compared with REF (dashed green line).

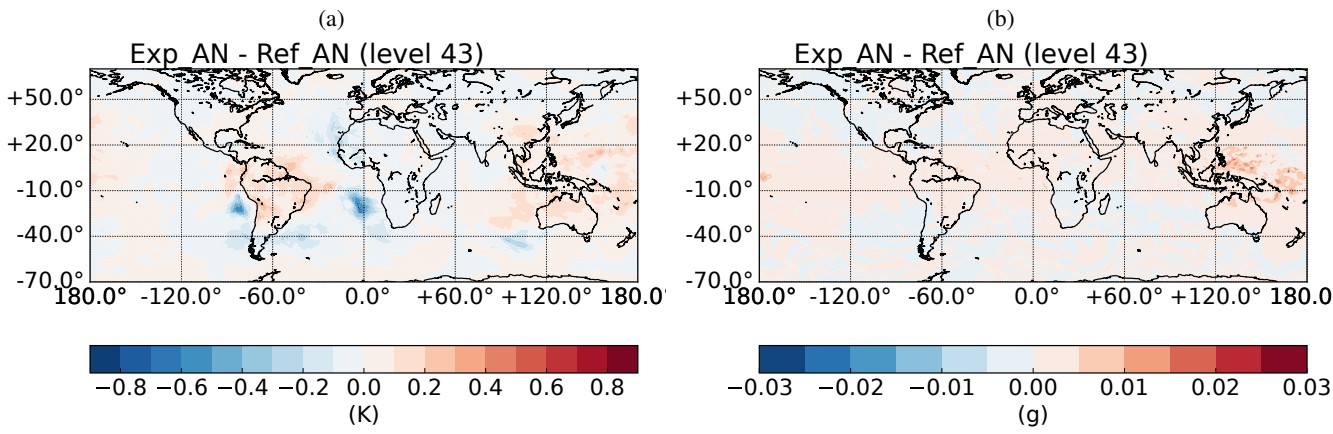

**Figure 7.** Temperature (a) and humidity (b) analyses difference between REF and EXP for the first assimilation cycle on 07 December 2017 at 00 UTC at ARPEGE model level 43 which corresponds to 200 hPa.

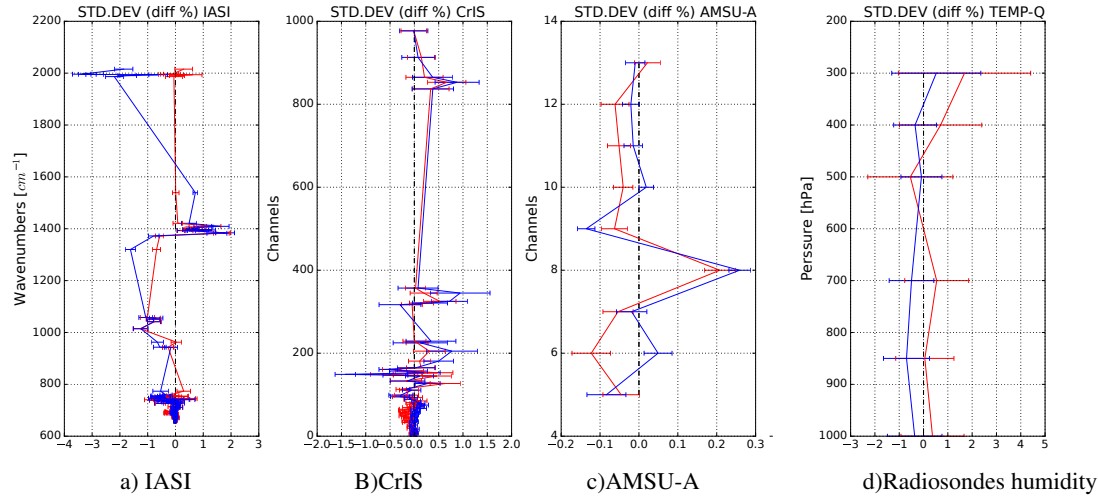

**Figure 8.** Relative differences of FG departure (red curve) and AN departure (blue curve) standard deviation between EXP and REF for IASI (a), CrIS (b), AMSU-A (c) and TEMP-q (d). The horizontal error-bars represent the 95% significance value for each difference.

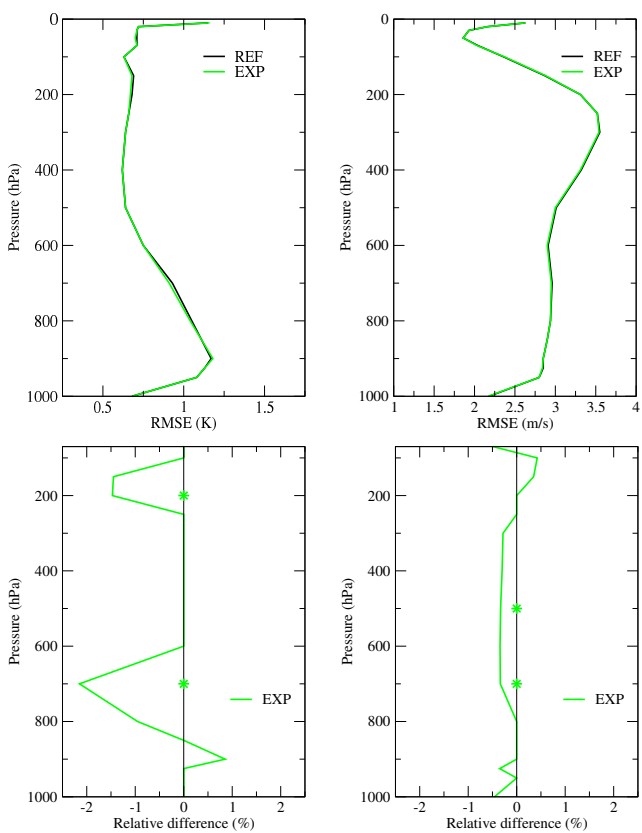

**Figure 9.** Root mean square error of the 12-hour forecast error for the Southern Hemisphere computed with respect to ECMWF analysis over the period from 7 December 2017 to 17 January 2018 for REF (in black) and EXP (in green) for temperature (left-hand side) and wind (right-hand side). The second line represents the relative difference with respect to the reference. Green stars indicate that the differences of EXP with REF are statisticaly significant according a Bootstrap test with a 99.5% confidence level.