# Peer review of "Homogeneity criteria from AVHRR information within IASI pixels in a Numerical Weather Prediction context"

_Atmospheric Measurement Techniques, 2018_

## Referee Comment (RC1) · Anonymous Referee #1 · 12 Dec 2018

This manuscript looks at an interesting area and some new results are presented but without getting into much depth of analysis. There are gaps to be filled to support the conclusions. In particular it is hard to make an informed choice between the different inhomogeneity screening options based on this work, as only one of the viable options was tested in an NWP system (the other tested option already showed clear defects even before NWP system testing). There is also no independent validation of whether the screening achieves its goal, which is homogeneous scenes.

Major points

1) P6 L9: "we plan to assimilate clear or cloudy observations that are completely covered in the IASI FOV .... discarding fractional cloud observations". This still allows the possibility of fully clear obs being assimilated in a fully cloudy model (or vice-versa). Is

that the intention? How do these screening methods treat cloudy modelled scenes if at all? The text should explain.

2) The choice of 49K^2 departure threshold in 3.2.2 is unsupported and uninvestigated in the text. Important questions are what this threshold means in terms of retained cloudy scenes, and how do its effects differ from those of the 7K AVHRR departure check in the Martinet et al. (2013) approach? Ultimately it should be investigated why the adapted (it is not the original) Eresmaa (2014) technique provides poor cloud screening here. Maybe it is the adaptation of this departure check? Finally, it is probably a case of poor wording rather than science, but it seems incorrect to claim the departure threshold as a check on homogeneity rather than just on cloud (P10 L19).

3) The choice of the 0.8% threshold on p11 is barely supported in the text or by Figure 2. It may be the colour scale but there seem to be no highly inhomogeneous scenes according to the IASI Ne (e.g. between 0.3 and 0.7 on Fig. 2) and there certainly seems no correlation between the relative cluster standard deviation and the Ne.

4) P11 L17 "Similarly in model space the D_mean..." Since D_mean is based on observation minus background it is not in model space and neither is it a direct indication of the model cloudiness.

5) "The percentage of cloudy AVHRR pixels" - if this is a good enough indicator of fractional cloud and/or inhomogeneity to use it to validate the screening criteria, why is it not used as part of the screening criteria?

6) Section 5, the intercomparison of selection criteria, does not fully make the case for the proposed selection method. M2013 keeps 29% of data in table 2 compared to 21% in the compromise method, with only slightly higher standard deviations. That could be a good choice, but it has been rejected at this stage. The balance between a slight increase in std. dev. and gaining extra data has hence not been properly explored. It seems odd to instead test the "Obs_HOM" approach in data assimilation as already from the intecomparison it is clear it does not work well. Further, without an exploration

of its sensitivity to threshold choices, the E2014 test does not have much of a chance in this intercomparison. Finally, each of the previous techniques M2013 and E2014, as well as the newly proposed compromise technique are composed of two tests, and it would be good to know how many rejections each is responsible for and how much overlap there is between the two tests.

7) Much of the conclusions will need to be updated to reflect a more thorough comparison of the different methods but particularly problematic is the statement P19 L4-6, saying that the M2013 and E2014 techniques were unsatisfactory due to a large loss of observations. Since M2013 provides more observations than the proposed method according to table 2, this statement is wrong. Also E2014 was not well explored in terms of thresholds and other possible adaptations to make it work in the current framework, using all-sky forward radiative transfer. Any rejection of this technique needs to be carefully qualified.

Minor points

1) Introduction: an up-to-date reference on homogeneity critetria for all-sky is the following. It should be discussed:

Okamoto, K. (2017), Evaluation of IR radiance simulation for all-sky assimilation of Himawari-8/AHI in a mesoscale NWP system. Q.J.R. Meteorol. Soc., 143: 1517-1527. doi:10.1002/qj.3022

2) P3 L31 "the first level at 10m" sounds odd; surely the lowest 10m of the atmosphere is also included in the model?

3) P5 L28 "Stratus Continental and Stratus Maritime" are cloud microphysical options in RTTOV; this should be stated; also it should be made clear how the choice is made, even if it is as obvious as using the land-sea mask.

4) P10 L1 "Background brightness temperature for AVHRR" - for clarity explain if this is clear-sky or all-sky.

5) P10 L7 If f^j is different from C_j please explain how; otherwise use consistent notation.

6) P11 L10 Instead of "relationship" is "ratio" intended? At present the text is imprecise.

7) P12 L15 "Mean standard deviation" is always a confusing phrase and needs explanation of what samples got meaned or standard deviated and in what order.

8) Table 2 should additionally include statistics for the observations that are fractionally cloudy according to AVHRR.

9) P12 L15 to P14 L10 is hard to read as it is overloaded with numerical results and bereft of much interpretation. Many of the numerical results are already listed in table 2 and do not need exhaustive restating in the text, which needs to be rewritten with a higher level of analysis for the reader. Where numerical results given in this text are not already in tables, they should be.

10) P14 L22 "Gaussian" is overly strong here and is not backed up by any statistical tests of Gaussianity.

11) P14 L30 "we keep more observations" - which method is referred to by "we"?

---

## Referee Comment (RC2) · Anonymous Referee #2 · 17 Dec 2018

General comments:

The English is generally very good, the paper is well structured and nicely concise. I am proposing major revisions because some of the plots should be remade, and the scope of the paper does not seem to fit the ambitions of the title. I found the choice of experiments a little strange, and was rather confused on the details. However, whilst I am suggesting some extensive changes, I think that these matters could be fixed quite easily by the authors.

I realised when I reached the final sections of this paper that I had totally misunderstood its intention. I had expected that the homogeneity criteria would be used to select additional observations, that are homogeneously cloudy, to assimilate in addition to the clear-sky channels accepted by the McNally and Watts check - i.e. to do something

similar to McNally (2009). It took me almost the whole paper to realised that what is being proposed is additional quality control on radiances already accepted by McNally and Watts. This does not seem like "preparation for all-sky assimilation" and I think that the scope of the paper should be revised.

At least, I assume that it is the case that this is just extra QC on clear sky calculations. . . I confess that I found the paper surprisingly confusing! There is no mention that RT-TOVCLD is being used in the assimilation expreiments, therefore I assume the calculations are clear sky, and there is no mention of the use even of a single grey-cloud layer scheme in use as in McNally (2009).

Some curious choices are made throughout the paper: I believe that the E2014 method was adapted because Eresmaa's intention was to keep only clear scenes, whereas you wish to allow through homogeneously cloudy scenes also, but in fact because you are rejecting observations that had already been allowed through by McNally and Watts, I don't see why you don't just apply Eresmaa's method without modification. Why do you not include the scheme of McNally (2009)? And finally, perhaps most surprisingly, you use the AVHRR clear/cloudy pixel fraction as a measure of whether the homogeneity criteria have "correctly" picked out homogeneous scenes, and on p15 you state that you are happy to accept a reasonable proportion of observations with >90% cloud cover. Why not just test the use of the AVHRR clear pixel fraction? And yet, you performed assimilation experiments with a scheme that you had seemed to reject based on the O-B statistics presented in Figure 4.

It is not really surprising that there is little impact, as very little seems to have changed in the experiments relative to the control. The work therefore seems rather immature for a publication. You still apply the CO2-slicing method (p16 line 17) - presumably this is designed to reject cloudy scenes? What effect does this have on the homogeneous scenes?

Specifics: P9 line 7: it is not clear whether this 7K check is an additional criterion

over the 8% criterion in the previous paragraph. Why 7K? I also don't understand "interpolated using 12 points" - is that 12 points in 3D?

P10 line 14-17: I don't understand how this first bullet point is different from the original method P10 line 18-19: Why 49K2? Other than that it fits the 7K applied to M2013, it seems quite high relative to 1K.

P11 line 5: This is the same as the first test of Martinet but with 2 channels. It would be clearer if this was stated. Why change the L to R in the equation? What does the addition of the second channel bring in practical terms?

P12: What is this dataset of 59 million observations? Is it just 24 hours' worth of observations? You state that 50% of the observations are 100% cloudy - that sounds potentially high for a normal dataset?

P12/Figure 3: Are the numbers in the text for bias and SD an average over a number of channels, or the maximum value from the windows? It should probably be the latter. I cannot match the figures in the text with the plots - the numbers do not seem to match (e.g. 11.7K bias -> the bias looks over 12 K in the figure). It would also be better to just plot Band 1 so we can see the effect on the temperature channels. I think the numbers scattered over several paragraphs and two pages would be better in a table.

P14 line 15-20: I would disagree that the distribution asymmetry is small. I also disagree that the Obs_HOM approach reduces the range of the tropospheric water vapour channel distribution.

P15: The discussion focuses on letting through the most data - this isn't necessarily the best criterion, as you may be letting through inhomogeneous scenes. There is trade off between more data and better data. M2013 and Obs_HOM let through a lot of partially cloudy scenes (and even 100% cloudy scenes may have different cloud types in one pixel).

P16: It is not clear what the set-up for the assimilation experiment is - you do not

mention RTTOVCLD - presumably this is still clear sky.

Figures: Figure 1: would be better as two bigger O-B plots. Figure 2: very strange Y-axis. You can't see much on these plots. Is the Y-axis expressed as % as in the criterion on p11 line 11? Figure 4: Why not plot a temperature sounding channel? The x-axis has strange divisions. It would be better symmetrical. Figure 6: I honestly cannot see any difference between these three plots. You need to revise the colour scale to highlight the differences. Figure 7: I cannot see the REF line: is it under the green line or the red line? This is an important figure as it is the first time I realised this paper was about improved QC (more obs are assimilated with Experiment B than A).

Tables: Table 2: Should include the % partially cloudy Tables A1 and A2 are unnecessary - this information is presumably included elsewhere. If not, a simple list of channel numbers would suffice.

Minor points:

P2 line 9: seems to be the first use of IR without the abbreviation being expanded. Section 2: this section is a light-touch description of the model and IASI, as it should be, but it is important to get the details correct in that case and make sure the writing is clear: P4 has a few poorly worded sentences, or poorly explained concepts. P4 Line 7 - the background error statistics are not "derived from a climatological matrix" - it isn't actually a matrix, and you do not explain how the ensemble information is incorporated. P4 Line 22 - this area needs rewriting - Presumably you mean that the accuracy of the forward model calculation is limited by the accuracy of the NWP model, and that for some variables this is not sufficient to correctly model the observations? "Modelisation" -> "Modelling" in English! P4 Line 29 - The McNally & Watts scheme is not clearly described. P4 Line 31: In this section, there are numerous references to CTOP and Ne, but suddenly you switch to PTOP - maybe Pangaud (2009) used PTOP instead of CTOP but this switch is not necessary.

P5 line 1: "IASI is a key element of the payload of the Metop series of European. . ." P5

para 1: you may as well update this with Metop-C launch date P5 line 22: It's a bit far to say that failing to assimilate cloudy IR observations is a source of error. P5 line 25: "allows to better describe" - not good english "allows a better desrciption of..." P5 line 29: You should reference the Baran parameterisation if you are going to mention it.

P6 line 8: "an innovative challenge"? Remove the word innovative. P6 line 8: The sentence "In the context of..." doesn't make sense. P6 line 21: THey are not IASI L1c products - they are components of the L1c product. P6 line 24: this sentence is not clear either. P6: line 28: this sentence is not clear. I think it is a stretch to say something with one class can be less homogeneous than something with multiple classes - this is a bit subjective.

P9 line 11: "aimed to propose" - that is a bit of a negative slant on this reference! "Proposed" would be better!

P17 line 15: No need to reference Table 3 here - it is a very basic table adn you describe it all in the text,
* * *

---

## Author Comment (AC1) · 6 Mar 2019

Reviewer 1 :

We thank Reviewer 1 for his/her comments which helped to improved, we hope, the quality of the manuscript. Reviewer 1's comments are in bold font, our answers are written with normal font.

**This manuscript looks at an interesting area and some new results are presented but without getting into much depth of analysis. There are gaps to be filled to support the conclusions. In particular it is hard to make an informed choice between the different inhomogeneity screening options based on this work, as only one of the viable options was tested in an NWP system (the other tested option already showed clear defects even before NWP system testing). There is also no independent validation of whether the screening achieves its goal, which is homogeneous scenes.**
An independent validation was added. The cloud homogeneity was compared with an homogeneity criterion based on cloud types retrieved from SEVIRI observations. We found similar results as those obtained with the AVHRR cloud cover. In addition we have removed one of the data assimilation experiment in order to keep only the experiment with the COMPR criteria.

**Major points**
**1) P6 L9: "we plan to assimilate clear or cloudy observations that are completely covered in the IASI FOV .... discarding fractional cloud observations". This still allows the possibility of fully clear obs being assimilated in a fully cloudy model (or vice-versa). Is that the intention? How do these screening methods treat cloudy modeled scenes if at all? The text should explain.** In the context of the preparation of all-sky assimilation, we plan to assimilate indiscriminately clear or cloudy observations that are completely covered in a homogeneous way, discarding the cases of fractional cloud observations : the clear cases would be assimilated as in the current operational version and the cloudy ones in manner to be determined. The comparison to modeled scenes is ensured with the background departure check. In this case it will not be possible to keep a clear observation with a fully cloudy model.

**2) The choice of 49K^2 departure threshold in 3.2.2 is unsupported and uninvestigated in the text. Important questions are what this threshold means in terms of retained cloudy scenes, and how do its effects differ from those of the 7K AVHRR departure check in the Martinet et al. (2013) approach? Ultimately it should be investigated why the adapted (it is not the original) Eresmaa (2014) technique provides poor cloud screening here. Maybe it is the adaptation of this departure check? Finally, it is probably a case of poor wording rather than science, but it seems incorrect to claim the departure threshold as a check on homogeneity rather than just on cloud (P10 L19).**
The choice of 49K^2 departure was studied with the graph of leaving observations as a function of the departure. As shown below, this threshold allows to keep more than 50 % of the observations. That is why this threshold was kept. In addition it fits the 7K AVHRR departure check of M2013.
It is proposed in the text the following sentence « is less than $49K^2$ . This particular value of threshold allows to keep more than 50 % of the observations compared to the initial threshold of $1K^2$ by Eresmaa (2014) which retains only 10 % of the observations. In addition this threshold compares well with the one applied by M2013, but it is applied over the 2 IR AVHRR channels. »
The compromise selection method is an adaptation of the E2014 and is strongly based on it as we used the 2 IR AVHRR channels and the distance $D_{mean}$ proposed by E2014 is used for the background check.

[Figure]

We agree that the $D_{mean}$ based check cannot be considered as a homogeneity check and the sentence (initially P10 l19) has changed : « In this test, we used the $D_{mean}$ proposed by Eresmaa (2014) to perform a kind of cloudiness consistency check between the observation and the model simulation »

**3) The choice of the 0.8% threshold on p11 is barely supported in the text or by Figure 2. It may be the colour scale but there seem to be no highly inhomogeneous scenes according to the IASI Ne (e.g. between 0.3 and 0.7 on Fig. 2) and there certainly seems no correlation between the relative cluster standard deviation and the Ne.**

We recognise that the original Figure 2 was not clear with the interpolation. Figure 2 has been plotted with another color scale, a logarithm scale on y axis and as a function of data count.

[Figure]

As can be seen in the Figure, the threshold of 0.8% allows to remove 39,8% and 42,0% of observations for each AVHRR channel.

**4) P11 L17 "Similarly in model space the D_mean..." Since D_mean is based on observation minus background it is not in model space and neither is it a direct indication of the model cloudiness.**

We agree that D_mean is computed in the observation space. We propose the following rewording : « Similarly, we used the background departure check in the observation space D_mean (presented in section 3.2.2).

**5) "The percentage of cloudy AVHRR pixels" - if this is a good enough indicator of fractional cloud and/or inhomogeneity to use it to validate the screening criteria, why is it not used as part of the screening criteria?** We agree that this percentage of cloudy AVHRR pixels is not so good indicator of the presence of homogeneous clouds to assess the screening criteria. As said above, we propose here an independent evaluation against SEVIRI cloud type. The results obtained with these data are similar to those obtained with the AVHRR cloud cover.

**6) Section 5, the intercomparison of selection criteria, does not fully make the case for the proposed selection method. M2013 keeps 29% of data in table 2 compared to 21% in the compromise method, with only slightly higher standard deviations. That could be a good choice, but it has been rejected at this stage. The balance between a slight increase in std. dev. and gaining extra data has hence not been properly explored. It seems odd to instead test the "Obs_HOM" approach in data assimilation as already from the intecomparison it is clear it does not work well. Further, without an exploration of its sensitivity to threshold choices, the E2014 test does not have much of a chance in this intercomparison. Finally, each of the previous techniques M2013 and E2014, as well as the newly proposed compromise technique are composed of two tests, and it would be good to know how many rejections each is responsible for and how much overlap there is between the two tests.**

We agree with the reviewer that the issue here is to find a compromise between good statistics and a sufficient number of observations for the assimilation. Here the Obs_HOM approach clearly keeps too much observations for the assimilation. In section 5, this experiment with the Obs_Hom Criteria was removed and we keep only the COMPR criteria for the data assimilation experiment. The COMPR method results from the E2014 as the D_mean criterion was used.

Please find below a table summarizing the impact of each criteria for each method.

| | % of remaining observations | When 2 criteria applied | When all criteria applied |
|---|---|---|---|
| **M2013** | | | |
| Sigma inter 1 | 89,8% | 84,8% | 53,9% |
| Sigma intra 1 | 87,8% | | |
| Background check 1 | 59,4% | | |
| **E2014** | | | |
| Crit 1 can 1 | 39,8% | 39,4% | 22,8% |
| Crit1 can 2 | 42,9% | | |
| Dmean<49K | 42,7% | | |
| **COMPR** | | | |
| obs_hom 1 | 68,2% | 67,3% | 36% |
| Obs hom 2 | 69,6% | | |
| Dmean<49K$^2$ | 42,7% | | |
| OBS_Hom 1+Dmean 1 | 43,3% | 36% | |
| obs_Hom 2+Dmean 2 | 44,7% | | |

As can be seen in the previous table, if tests based on a single channel allow to keep more or less the same percentage of observations, the combination of both channels lead to a more restrictive choice.

**7) Much of the conclusions will need to be updated to reflect a more thorough comparison of the different methods but particularly problematic is the statement P19 L4-6, saying that the M2013 and E2014 techniques were unsatisfactory due to a large loss of observations. Since M2013 provides more observations than the proposed method according to table 2, this statement is wrong. Also E2014 was not well explored in terms of thresholds and other possible adaptations to make it work in the current framework, using all-sky forward radiative transfer. Any rejection of this technique needs to be carefully qualified.**

We have removed the sentence. The conclusions of this study were updated.

A new method using of collocated AVHRR cluster information to improve the selection of homogeneous IASI observation scenes within the numerical weather prediction ARPEGE model has been developed at Météo-France for data assimilation purposes and has been presented in this study.

The first step consisted in adapting the IASI observation operator based on the RTTOV radiative transfer model by using the RTTOV-CLD module with cloudy microphysical parameters (liquid water content (ql), ice content (qi) and cloud fraction) for the simulation of cloudy radiances. A qualitative evaluation of such module showed realistic simulated cloud structures at various locations around the globe with a quite good agreement against IASI observations.

The second and main step of this work was to assess the impact of several methods used to select homogeneous IASI observations using AVHRR clusters. Two selection methods (derived from the literature : Martinet et al. (2013) and Eresmaa (2014))) were preliminarily evaluated. Despite a good improvement in terms of biases and standard deviations of the FG departures, it was found that these two methods were not satisfactory in an operational context (in assimilation) to a large IASI observation reduction. the criteria from the Martinet et al. (2013)'s method favours the homogeneous cloudy observations and retains more than a half of the observations while the Eresmaa (2014)'s method gives priority to clear observations and keeps only 22% of the observations. Then, two new sets of criteria were defined from these two previous methods in order to have a more balanced choice of clear and cloudy observations and good statistics in terms of bakcground departures and implemented within the ARPEGE model :

– The first criterion derived from Martinet et al. (2013) method looks for the consistency between different clusters occupying the same IASI FOV by examining this homogeneity relative to the weighted average brightness temperature of the AVHRR clusters; it is only based on observations and computed for both infrared AVHRR channels as in Eresmaa (2014). This criterion allows to retain 67% of observations.

– In addition, the second criterion is derived from Eresmaa (2014)'s test and assesses the coherence of each cluster compared to the background brightness temperature simulation ; it is in fact a good compromise between the previous criterion and the two "historical" ones with accurate statistics and a sufficient number of observations 36% that passed the check. It also allows to retain the same proportion of homogeneous clear and cloudy observations contrary to the derived Martinet et al. (2013) and Eresmaa (2014) methods.

Therefore, assimilation experiments were conducted to assess the impact of these new selecting homogeneous IASI observation features in the current clear sky assimilation. This revised check was added to the McNally and Watts (2003) cloud detection. The results obtained in this case show that the scenes categorization has been facilitated and cloudy observations can be better filtered out compared to what is done in the operational ARPEGE version. 3% of all observations are rejected with the compromise method for the assimilation. The impacts on the first guess and analysis departures (showing more Gaussian shape) are generally low but with a beneficial reduction on the standard deviation of first guess departures mainly on the IASI and AMSU-A observations. Regarding the forecasts scores, neutral impact is reported when these selection criteria are taken into account on top of the McNally and Watts (2003) algorithm.

**Minor points**

**1) Introduction: an up-to-date reference on homogeneity critetria for all-sky is the following. It should be discussed: Okamoto, K. (2017), Evaluation of IR radiance simulation for all-sky assimilation of Himawari-8/AHI in a mesoscale NWP system. Q.J.R. Meteorol. Soc., 143: 1517-1527. doi:10.1002/qj.3022** A reference to the work by Okamoto (2017) was added in the text, after the section on the selection of cloudy scenes based on cloud homogeneity. « Okamoto (2017) studied the impact of the super-observation homogeneity quality control on the Advanced Himawari Imager brightness temperature simulation. He concluded that for larger size of super-observations, the standard deviation threshold should be relaxed in order to keep sufficiently low brightness temperatures associated with high-level cloud because of the presence of more cloud heterogeneity in large size observations»

**2) P3 L31 "the first level at 10m" sounds odd; surely the lowest 10m of the atmosphere is also included in the model?** The first level of the ARPEGE is well at 10m. Fields below this level are interpolated from the model and the surface.

**3) P5 L28 "Stratus Continental and Stratus Maritime" are cloud microphysical options in RTTOV; this should be stated; also it should be made clear how the choice is made, even if it is as obvious as using the land-sea mask.** The cloud type chosen depends on the land-sea mask of the model. Over land, the cloud type stratus continental is chosen, the stratus maritime is used over sea. The sentence was reworded as follows : « To simulate the radiances observed in cloudy conditions using RTTOV-CLD, we use two main cloud types : firstly liquid water cloud which corresponds to two RTTOV-CLD cloud microphysical options depending on the land sea mask of the model (Stratus continental over land and Stratus Maritime over the sea),  secondly the ice water cloud of the Cirrus type, using Baran parameterisation (Baran et al 2014 and Vidot et al 2015) to define the optical properties

**4) P10 L1 "Background brightness temperature for AVHRR" - for clarity explain if this is clear-sky or all-sky.** Here this background brightness temperature is simulated with clear sky as in Eresmaa (2014). This was clarified in the text « where $R_i^{BG}$ is the clear-sky background brightness temperature for AVHRR channel i ».

**5) P10 L7 If fˆj is different from C_j please explain how; otherwise use consistent notation.**
$f^j$ and $C_j$ are the same quantity, the fractional coverage of the cluster. $f_j$ was changed into  $C_j$.

**6) P11 L10 Instead of "relationship" is "ratio" intended? At present the text is imprecise.** Relation ship was changed into ratio.

**7) P12 L15 "Mean standard deviation" is always a confusing phrase and needs explanation of what samples got meant or standard deviated and in what order.** The paragraph was rewritten and there is no more mean standard deviations.

**8) Table 2 should additionally include statistics for the observations that are fractionally cloudy according to AVHRR.** This column was added in Table 2.

**9) P12 L15 to P14 L10 is hard to read as it is overloaded with numerical results and bereft of much interpretation. Many of the numerical results are already listed in table 2 and do not need exhaustive restating in the text, which needs to be rewritten with a higher level of analysis for the reader. Where numerical results given in this text are not already in tables, they should be.**

This part of the paper was rewritten and numbers scattered along the text were removed. "The percentage of cloudy AVHRR pixels in the IASI field was also used to assess the choice of homogeneity criteria (Table 3).

Our global data set is made of 50% of the observations entirely covered by clouds and 12% of clear observations according to the AVHRR cloud cover. These results obtained over the globe set agree well with the ones obtained with SEVIRI data over the Atlantic Ocean. Except for M2013 method, the percentage of selected observations for each selection method is larger (+2/4 %) with the SEVIRI data evaluation than with the AVHRR cloud cover. The bias and standard deviation of observations minus simulations (O-G), are shown in Figure 3.(a) for the 314 IASI channels. As expected, the best statistics are obtained for channels less affected by clouds (e. g. $CO_2$ and water vapour high peaking channels).

For the whole dataset, window channels present a bias of around -0.6K. The standard deviations are larger (around 12 K) for window channels sensitive to the surface. With the M2013 selection method (figure 3. (b)), the standard deviation of window channels is reduced to around 4 K as well as the bias close to zero. The standard deviation of the other channels (680-780cm$^{-1}$ is also well decreased. The E2014 selection method (figure 3. (c)) improves the bias and the standard deviation (2.0 K for window channels) for all the channels. As expected, the impact is larger for surface sensitive (and thus cloud sensitive) channels than for the tropospheric channels (680-780cm$^{-1}$). On the contrary, with the Obs_HOM method (figure 3. (d)), small statistics improvement is obtained for the standard deviation and the bias. The statistics obtained with the COMPR method (Fig. 3.(e)) are reduced compared to the whole data set and slightly less good than with the initial E2014 method (for window channels the standard deviation is around 2.2 K instead of 2 K for E2014)."

**10) P14 L22 "Gaussian" is overly strong here and is not backed up by any statistical tests of Gaussianity.** We propose the following modification « the O-G distributions become symmetrical and get closer to the gaussian distribution »

**11) P14 L30 "we keep more observations" - which method is referred to by "we"?** This corresponds to E2014 method and it was changed in the text.

---

## Author Comment (AC2) · 6 Mar 2019

Response to Reviewer 2.

We thank Reviewer 2 for his/her comments which helped to improved, we hope, the quality of the manuscript. Reviewer 2's comments are in bold font, our answers are written with normal font.

**General comments:**
**The English is generally very good, the paper is well structured and nicely concise. I am proposing major revisions because some of the plots should be remade, and the scope of the paper does not seem to fit the ambitions of the title. I found the choice of experiments a little strange, and was rather confused on the details. However, whilst I am suggesting some extensive changes, I think that these matters could be fixed quite easily by the authors.**
To agree with the content of the paper, we suggest to change the title of the paper with « Homogeneity criteria from AVHRR information with IASI pixels in a Numerical Weather Prediction context ». In addition we have removed the data assimilation experiment with the only Obs-Hom criteria and only kept the experiment with the compromise criteria. Some parts of section 4 were rewritten as well as the conclusions.

**I realised when I reached the final sections of this paper that I had totally misunderstood its intention. I had expected that the homogeneity criteria would be used to select additional observations, that are homogeneously cloudy, to assimilate in addition to the clear-sky channels accepted by the McNally and Watts check - i.e. to do something similar to McNally (2009). It took me almost the whole paper to realised that what is being proposed is additional quality control on radiances already accepted by McNally and Watts. This does not seem like "preparation for all-sky assimilation" and I think that the scope of the paper should be revised.**
We recognised that the title of the paper may be confusing that is why we propose to change the title of the paper with « Homogeneity criteria from AVHRR information within IASI pixels in a Numerical Weather Prediction context » The objective of the paper in the last paragraph of the introduction was also modified into : « Our objective is to determine homogeneity criteria valid for both clear and cloudy conditions, suitable to an NWP context using collocated AVHRR and IASI information. »

**At least, I assume that it is the case that this is just extra QC on clear sky calculations**
**I confess that I found the paper surprisingly confusing! There is no mention that RTTOV-CLD is being used in the assimilation experiments, therefore I assume the calculations are clear sky, and there is no mention of the use even of a single grey-cloud layer scheme in use as in McNally (2009).**
We propose an ensemble of homogeneity criteria. In case of clear sky, the McNally and Watts is applied as an additional QC test afterwards. In case of cloudy scenes, these criteria could pave the way to an all-sky assimilation (methodology not decided yet). You are right in the assimilation experiments, only RTTOV was used (clear sky assimilation). RTTOVCLD was only used to compute the homogeneity criteria based on cloudy AHVRR simulations. In the operational version of ARPEGE , a single layer grey cloud scheme is used (Guidard et al 2011) but in the experiments carried out in this paper, this possibility has been switched off to focus on clear sky assimilation.
This has been specified in the text : « In these experiments, no cloudy observations detected with the CO2-slicing method and used with a single grey-cloud layer scheme was assimilated unlike in Guidard et al. (2011). RTTOVCLD was only used to compute the homogeneity criteria based on cloudy AHVRR simulations and RTTOV was used for the clear sky assimilation. »

**Some curious choices are made throughout the paper: I believe that the E2014 method was adapted because Eresmaa's intention was to keep only clear scenes, whereas you wish to allow through homogeneously cloudy scenes also, but in fact because you are rejecting observations that had already been allowed through by McNally and Watts, I don't see why you don't just apply Eresmaa's method without modification. Why do you not include the scheme of McNally (2009)?**
The objective of Eresmaa (2014) method is to keep clear pixels. Our objective is to keep homogeneous scenes both in clear and cloudy conditions. The Eresmaa method is not suitable in our case and this why we propose the modification based on this method. The single layer grey

cloud scheme could have been used but the selection of cases to be assimilated this way woud need a dedicated study.

**And finally, perhaps most surprisingly, you use the AVHRR clear/cloudy pixel fraction as a measure of whether the homogeneity criteria have "correctly" picked out homogeneous scenes, and on p15 you state that you are happy to accept a reasonable proportion of observations with >90% cloud cover. Why not just test the use of the AVHRR clear pixel fraction? And yet, you performed assimilation experiments with a scheme that you had seemed to reject based on the O-B statistics presented in Figure 4.**
As suggested by reviewer 1 we added an evaluation with independent data of cloud type from SEVIRI on board MSG satellite in section 4 because the cloud cover from the AVHRR has some defects. We agree that observations with >90 % cloud cover are not necessarily homogeneous.
We have now removed the data assimilation experiments with obs_HOM criteria and we kept only the                                                        compromise                                                        method.

**It is not really surprising that there is little impact, as very little seems to have changed in the experiments relative to the control. The work therefore seems rather immature for a publication. You still apply the CO2-slicing method (p16 line 17) - presumably this is designed to reject cloudy scenes? What effect does this have on the homogeneous scenes?**
The sentence p16 l 17 was confusing and thus modified as said above : «  In these experiments, no cloudy observations detected with the CO2-slicing method and used with a single grey-cloud layer scheme was assimilated unlike Guidard et al. (2011). RTTOVCLD was only used to compute the homogeneity criteria based on cloudy AHVRR simulations and RTTOV was used for the clear sky assimilation. »
 3 % of observations are rejected between Exp and Ref, so you are correct that it is not very surprising that there is very little impact. Nevertheless, this impact is slightly positive. Thus it seems that this method is reliable enough to be used in a NWP assimilation. These criteria will reveal their full potential in an all-sky assimiulation. The design of a IR all-sky assimilation has still to be done.

**Specifics:**
**P9 line 7: it is not clear whether this 7K check is an additional criterion over the 8% criterion in the previous paragraph. Why 7K? I also don't understand "interpolated using 12 points" - is that 12 points in 3D?** This 7K check is an additional check verifying that both the observation and the model observe the same cloudy scene. In the original Martinet et al (2013) it was applied to the difference between the mean AVHRR brightness temperatures from the observed and simulated clusters. In the case or ARPEGE the horizontal mesh is coarser and we replaced this fine scale check with the difference between the brightness temperature from the guess profile and the observation. The guess profile results from a horizontal interpolation of 12 profiles surrounding the observation position coming from a 6-hour forecast.
We have added this point in the text : « In the original Martinet et al (2013) study, a third check verifying that both the observation and the model observe the same cloudy scene was done with the difference between the mean AVHRR brightness temperatures from the observed and simulated clusters less than 7 K. Here, the ARPEGE model has a coarser resolution and it is not possible to simulate the AVHRR clusters. This check was adapted with the difference between the AVHRR observation and the AVHRR simulation from the guess, which come from a horizontal interpolation of the 12 profiles surrounding the observation position coming from a 6-hour forecast. »

**P10 line 14-17: I don't understand how this first bullet point is different from the original method.** The difference lies in the fact that the simulation computation were done with RTTOV-CLD. This is mentioned now in the text. « All AVHRR simulations from background are made with RTTOV-CLD and the threshold of the background departure check was modified » The description of the threshold for inter-cluster homogeneity was thus removed.

**P10 line 18-19: Why 49K2? Other than that it fits the 7K applied to M2013, it seems quite high relative to 1K.**
Many trials were done for the value of $D_{mean}$. Below you will find the function of leaving observations as a fonction of $D_{mean}$. It appears that this value allows to keep more than 50 % of the observations. In addition it also fits the 7K threshold of M2013 but over the 2 IR AVHRR channels.

[Figure]

We used the D mean proposed by Eresmaa (2014) to perform here a kind of cloudiness consistency check between the observation and the model simulation if D mean is less than 49 $K^2$. This particular value of threshold allows to keep more than 50% of the observations compared to the initial threshold of 1$K^2$ by Eresmaa (2014) which retains only 18% of the observations. In addition, this threshold compares well with the one applied by M2013, but it is applied over the 2 IR AVHRR channels. The text was modified accordingly.

**P11 line 5: This is the same as the first test of Martinet but with 2 channels. It would be clearer if this was stated. Why change the L to R in the equation? What does the addition of the second channel bring in practical terms?**
It was done : « It is the same test as in M2013 but in the brightness temperature space ». The change of L into R is practical as it is easier to work with brightness temperatures. We chose to have all value in brightness temperatures. If we only consider channel 1, we keep 68,2 % of the observations, with channel 2 69,6 % are remaining but with both 67,3 % of the observations are selected. This was mentioned in the text : « If this test is only applied over channel (10:5 m, 68,2% of the observations are selected, if applied over channel 11:5 m), 69,6% of the data are kept and if it is applied over both channels, 67,3% pass the test. »

**P12: What is this dataset of 59 million observations? Is it just 24 hours' worth of observations? You state that 50% of the observations are 100% cloudy - that sounds potentially high for a normal dataset?** There was a confusion between the number of observations and the number of channels. In fact the statistics were computed over 188090 IASI observations for 30 January 2017 and the number was corrected in the text and in the table. Half of the observations having a cloud cover of 100 % may seem a very high percentage, but with the independent validation we proposed with SEVIRI data, we find the same result.

**P12/Figure 3: Are the numbers in the text for bias and SD an average over a number of channels, or the maximum value from the windows? It should probably be the latter. I cannot match the figures in the text with the plots - the numbers do not seem to match (e.g. 11.7K bias -> the bias looks over 12 K in the figure). It would also be better to just plot Band 1 so we can see the effect on the temperature channels. I think the numbers scattered over several paragraphs and two pages would be better in a table.**

Figure 3 was changed and represents now only the spectral range of band 1. Section 4 was partly rewritten. This section was re-arranged and there are now less numbers in the text. Tables were also modified in order to show statistics on window channels and CO2 channels.

**P14 line 15-20: I would disagree that the distribution asymmetry is small. I also disagree that the Obs_HOM approach reduces the range of the tropospheric water vapour channel distribution.** Both sentences were removed and changed with. « The distribution asymmetry is reduced for mid and low tropospheric water vapour channels with M2013 and E2014 selection. » for the first one.

**P15: The discussion focuses on letting through the most data - this isn't necessarily the best criterion, as you may be letting through inhomogeneous scenes. There is trade off between more data and better data. M2013 and Obs_HOM let through a lot of partially cloudy scenes (and even 100% cloudy scenes may have different cloud types in one pixel).** We agree that 100% cloudy scenes may have different cloud types in one pixel. Now the discussion is based on SEVIRI cloud which is an independent validation datum. We also agree that there is a trade-off between more data provided by M2013 and Obs_HOM and better data. That is why we propose a fourth method based on the Eresmaa (2014) test.

**P16: It is not clear what the set-up for the assimilation experiment is - you do not mention RTTOVCLD - presumably this is still clear sky.**
Now it is clearly stated : « RTTOVCLD was only used to compute the homogeneity criteria based on cloudy AHVRR simulations and RTTOV was used for the clear sky assimilation. »

**Figures:**
- **Figure 1: would be better as two bigger O-B plots.** We have replaced the simulation plots with the suggested O-B plots. The comments on Figure 1 have been changed accordingly :

b) Observations minus RTTOV (clear-sky) simulation

c) Observations minus RTTOV-CLD simulations

[Figure]

« To illustrate the benefit brought by RTTOV-CLD, Figure (1) shows IASI brightness temperature observations of a cloud-sensitive surface channel (1271, 962.5 cm −1 ) and differences between the observations and the simulations computed with RTTOV considering clear-sky and with RTTOV-CLD. Brightness temperatures less than 250 K are usually associated with higher elevation cloud structures. By using RTTOV in clear sky (figure 1.b) to simulate IASI observations, the observation departures are mainly below zero and may reach up to -60 K. This can be

explained by the fact that the main cloud structures associated with low values of brightness temperature for the surface channel are missing in the simulation. On the contrary, differences obtained with the RTTOV-CLD simulations are in overall in better agreement with lower positive and negative values (figure 1.c). No major differences are found for example for the cloud structures located over the North Atlantic (30N-70N, 40W-0W) and above (30S-70S, 60W-0W) the Southern Atlantic Ocean. Large difference values are mainly obtained in the Tropics region. This may be explained by the fact that clouds are better simulated in the ARPEGE model for mid-latitudes than in the Tropics. »

- **Figure 2: very strange Y-axis. You can't see much on these plots. Is the Y-axis expressed as % as in the criterion on p11 line 11?** Yes the Y-axis is expressed as % as in the criterion line 10 page 11. The figure changed, with a log scale on the y axis, another color scale and as a function of data count.

[Figure]

- **Figure 4: Why not plot a temperature sounding channel? The x-axis has strange divisions. It would be better symmetrical.** Panels of Figure 4 were drawn again in order to have symetrical x axis as shown below. We do not consider temperature sounding channel as we are more interested in humidity and clouds.

[Figure]

**Figure 4.** Frequency distribution of brightness temperature difference between observation and background (O-G) for all observations (a, b, c), after applying the homogeneity criteria derived from Martinet et al 2013 (d, e, f), the homogeneity criteria derived from Eresmaa 2014 (g, h, i,), the third method based on observation space method (j, k, l) and the compromised approach (m, n, o). The PDF are presented for three channels: window channel 1271, low-tropospheric water vapour channel 5403, and mid-tropospheric water vapour channel 2701). The Gaussian distributions with the same error characteristics (mean and standard deviation) are also shown in blue dashed lines.

- **Figure 6: I honestly cannot see any difference between these three plots. You need to revise the colour scale to highlight the differences.**
  As the figure is a bit redundant with figure 7 we propose to remove it.

- **Figure 7: I cannot see the REF line: is it under the green line or the red line? This is an important figure as it is the first time I realised this paper was about improved QC (more obs are assimilated with Experiment B than A).**

[Figure]

Figure 7 (a) has been changed with this upper-panel representing the number of assimilated data in REF and the second panel represent the the relative difference of number of observations for EXP (EXP.B) compared with REF (EXP.A being removed from this section). The caption was changed accordingly.

**Tables:**
- **Table 2: Should include the % partially cloudy**
  The column of partially cloudy observations was added in Table 2.
- **Tables A1 and A2 are unnecessary - this information is presumably included elsewhere. If not, a simple list of channel numbers would suffice.**
  Tables were removed.

**Minor points:**
- **P2 line 9: seems to be the first use of IR without the abbreviation being expanded.**
  Done
- **Section 2: this section is a light-touch description of the model and IASI, as it should be, but it is important to get the details correct in that case and make sure the writing is clear: P4 has a few poorly worded sentences, or poorly explained concepts.**
  - **P4 Line 7 - the background error statistics are not "derived from a climatological matrix"- it isn't actually a matrix, and you do not explain how the ensemble information is incorporated.** We agree that this sentence was not well written and we replaced it with « The background errors are computed at each analysis time based on the 25-member assimilation ensemble (see Berre et al 2015 for further details). »

  - **P4 Line 22 - this area needs rewriting - Presumably you mean that the accuracy of the forward model calculation is limited by the accuracy of the NWP model, and that for some variables this is not sufficient to correctly model the observations?** This paragraph was rewritten : The observation operator allows to simulate observations from the model variables for comparison with the actual measurements. For satellites radiances, it includes a radiative transfer model.

The accuracy of the forward model calculation could be limited by the accuracy of the NWP model, for some variables this is not sufficient to correctly model the observations and these observations have to be discarded.

- **"Modelisation" -> "Modelling" in English!** Corrected.

- **P4 Line 29 - The McNally & Watts scheme is not clearly described. This scheme is now better described in the text :** The McNally and Watts (2003) scheme intends to detect clear channels and to assimilate channels unaffected by clouds even in a cloud-affected pixel. The channels are first re-ordered according to a ranking with respect to the altitude that reflects their relative sensitivity to the presence of cloud. After having applied a low-pass filter a search for the channel at which a monotonically growing departure can first be identified. Having found this channel all channels ranked more sensitive are flagged as cloudy and those ranked less sensitive are flagged clear.

- **P4 Line 31: In this section, there are numerous references to CTOP and Ne, but suddenly you switch to PTOP - maybe Pangaud (2009) used PTOP instead of CTOP but this switch is not necessary.** PTOP was modified into CTOP.

- **P5 line 1: "IASI is a key element of the payload of the Metop series of European: : :" P5 para 1: you may as well update this with Metop-C launch date.** The Metop-C launch data has been specified in then text: November 2018. « The third instrument was mounted on the Metop-C satellite, which was launched in November 2018. »

- **P5 line 22: It's a bit far to say that failing to assimilate cloudy IR observations is a source of error.** This sentence was modified : "Assimilation of cloudy radiances is a crucial challenge for NWP centres as the cloudy observations discard represent an under-exploitation of hyperspectral sounders especially in sensitive meteorological areas (McNally, 2002; Fourrié and Rabier, 2004)."

- **P5 line 25: "allows to better describe" - not good english "allows a better desrcription of: : :"** Change made.

- **P5 line 29: You should reference the Baran parameterisation if you are going to mention it.** The reference to Baran, A. J., Cotton, R., Furtado, K., Havemann, S. , Labonnote, L.-C., Marenco, F., Smith, A. and Thelen, J.-C. 2014: A self-consistent scattering model for cirrus. II: The high and low frequencies. Q.J.Roy. Meteorol. Soc., 140: 1039–1057. doi:10.1002/qj.2193
- **P6 line 8: "an innovative challenge"? Remove the word innovative.** Done

- **P6 line 8: The sentence "In the context of..." doesn't make sense.** The sentence was correted : « In the context of the preparation of all-sky assimilation, we plan to assimilate clear and cloudy observations that are completely covered in a homogeneous way, discarding the cases of fractional cloud observations. »

- **P6 line 21: They are not IASI L1c products - they are components of the L1c product.** This was corrected.

- **P6 line 24: this sentence is not clear either.** It has been reworded: For each class and each AVHRR channel, the cluster product provides the coverage of the class within the IASI pixel, the mean and the standard deviation of AVHRR brightness temperatures within the class.

- **P6: line 28: this sentence is not clear. I think it is a stretch to say something with one class can be less homogeneous than something with multiple classes - this is a bit subjective.** Our thought was that an important parameter is the standard deviation inside each class. The sentence was rewritten : A IASI FOV with several

classes, each one having a small standard  deviation and a mean radiance close to the ones of the other classes, can thus be more homogeneous than a FOV with a single class but with very large value of standard deviations.

- **P9 line 11: "aimed to propose" - that is a bit of a negative slant on this reference! "Proposed" would be better!** The change was made.

- **P17 line 15: No need to reference Table 3 here - it is a very basic table and you describe it all in the text,** The Table and the reference were removed.

---

## Author Response (AR2)

Dear Associate Editor,

Please find below how we have responded to the remarks of referee 1. We have also tried to improve the quality of the English of the paper. We hope that with all the modifications brought to the article, it could be now acceptable for publication.

Best regards,

Nadia Fourrié

**Referee 1**

**The manuscript has been significantly improved. The work now makes sense and follows a logical structure, with reasonable support for the conclusions. There is only one main issue (the quality of English) which is not the responsibility of the scientific reviewers. Assuming this will be dealt with, if a few other minor changes can be addressed the manuscript will be acceptable for publication.**

We thank Reviewer 1 for his/her comments which helped to improve, we hope, the quality of the manuscript. Reviewer 1's comments are in bold font, our answers are written with normal font.

**Main**

**The quality of English needs improving, particularly in the first few sections, either by the authors or by copy editing. Some examples are listed below but there are many more.**

We recognize that we are not native English speaking people. We tried to improve the quality of the English as you can see throughout the modifications we made in the text. We also took into account all the examples provided by the referee 1.

**Minor**

**P3 L10 - "Intuitively, collocated AVHRR data provide information on surface properties…" Please remove "Intuitively" since the following statements are unarguable.**

It was removed.

**P3 L18 - Sentence including "background equivalents to AROME fields" does not really make sense. Background AROME fields are being used to simulate cloud-affected equivalents to the observations.** You are right, the sentence was rewritten "This study was done in a 1D-Var framework using an advanced radiative transfer model (RTTOV-CLD) including profiles for liquid water content, ice water content and cloud fraction to simulate cloud-affected equivalents from background AROME fields."

**P9 L 11 - Section title "Background departure check" is not an accurate description here: it is a comparison of standard deviations, not O-B.** The referee is right and the title of the paragraph was modified with "Interclass homogeneity of the simulated cluster".

**P11 L29-31 - "we decided to select … less than 0.8%" - why specifically 0.8%? Please explain in the text a little more, as it is still not entirely obvious why here.** More explanations are now provided in the text: "This threshold allows to discard the population of observations with a large cloud cover and a large standard deviation ratio on the top right of the panels. It also allows to remove some observations for which the CO2-slicing algorithm has failed to retrieve a cloud top pressure and for which IASI cloud fraction is set to zero."

**Table 3 columns 3 and 4 could be swapped for consistency with table 2.** It was done

**Section 5.1: please recap exactly how IASI observations are used in the control and in the experiment as it is still not clear (covering the McNally and Watts and the new test and anything else important) To understand Fig. 6, we also need to know if there is a channel set for which the homogeneity criteria are not applied, due to the lack of influence of clouds on these channels.** More explanations on the application of the COMPR method are added in paragraphe 5.1: **"**In a second experiment called (EXP), we applied our COMPR approach (presented in the 3.2.4) on top of the Mc Nally and Watts cloud detection. As in Eresmaa (2014), these homogeneity criteria are provided to the McNally and Watts detection scheme and applied in its quick-exit scenario. This means that if the COMPR approach flags a homogeneous observation it can accelerate the decision of flagging the pixel as clear, but if the COMPR approach flags the observation as heterogeneous, the assimilation entirely relies on the McNally and Watts cloud detection scheme to discriminate which channels to assimilate. There is no specific channel set for which the homogeneity criteria are applied."

**P20 L29 -"results show" -> "results suggest" since the results from section 5 do not directly address the question of whether cloudy observations are better filtered, although clearly the reduction in IASI FG departure std. dev. could have been caused by this.** We agree and it was modified.

**P20 L 31 - "3% of all observations are rejected" needs more context - what percentage of all IASI observations are kept in the control and in the experiment? Please add the required results to section 5 if these are not already present.** We have modified the conclusion concerning the number of rejected observation compared to the reference. In overall, only around 19% of observations are kept for the assimilation. We have modified section 5.2 to add this information: "This proportion represents around 19.2% of the total amount of IASI observations available for the assimilation."
The last paragraph of page 20 was corrected. In average 1% of assimilated observations in reference are rejected in the experiment. "1% of assimilated observation in the reference are rejected with the homogeneity criteria. Depending on the spectral band, up to 15% of the number of total channels can be discarded with the use of the homogeneity criteria in the assimilation. The number of channels peaking high in the atmosphere (i. e. stratosphere) is of course not impacted by the homogeneity criteria, as the McNally and Watts algorithm always identifies them as clear."

**P21 L1 - "neutral impact" - actually Figs. 8a and 9 mainly suggest improvements to short range forecasts, which is worth mentioning. But what about longer range forecasts, please could you say something on this.** Regarding the forecasts scores, a very small positive impact at the 12-h forecast range for temperature and wind in the Southern Hemisphere when these selection criteria are taken into account on top of the McNally and Watts (2003) algorithm. However, at longer ranges, neutral impact is found.

Typos
**P4 L7 - "according a following terrain" -> "with a terrain-following"** Done

**P4 L18 - "divervence"** corrected

**P4 L31 - "allows to simulate" -> "allows simulation of"** Corrected

**P6 L16 - "are found for example for the" - rewrite** No major differences are found in the cloud structures as those present over the North Atlantic (30N-70N, 40W-0W) and above the Southern

Atlantic Ocean  (30S-70S, 60W-0W). They often consist in an alternation of positive and negative values suggesting a misplacement of the cloud structures.

**P8 L12 - "A IASI" -> "An IASI"** The 2 occurrences were changed.

**P8 L19 - "The first two ones" - remove "ones" here and later in the same sentence.** Done

**P8 L 27 - "determined by C_j WHICH is the cluster fraction"** Done

**P9 L1 - "calculed" -> "calculated"** Corrected

**P10 L3 - "If a IASI pixel do not" - use "an", "does"** Corrected

**P9 L9 - "The distance of each cluster .." is not a full sentence as lacking a verb.** The previous sentence was rewritten: "The intercluster consistency check relies on the comparison between the properties of the different clusters within the IASI FOV, the distance between each pair of clusters as well as the distance of each cluster to the background in both infrared AVHRR channels."

**P20 L16 - "bakcground"** It was modified.